# Liver group 2 innate lymphoid cells regulate blood glucose levels through IL-13 signaling and suppression of gluconeogenesis

Masanori Fujimoto[1,2], Masataka Yokoyama[1], Masahiro Kiuchi[3], Hiroyuki Hosokawa [4], Akitoshi Nakayama[1], Naoko Hashimoto[1], Ikki Sakuma[1], Hidekazu Nagano[1], Kazuyuki Yamagata [1], Fujimi Kudo[5], Ichiro Manabe [5], Eunyoung Lee[6], Ryo Hatano[6], Atsushi Onodera[3,7], Kiyoshi Hirahara [3], Koutaro Yokote[2], Takashi Miki [6,8], Toshinori Nakayama [3,9] & Tomoaki Tanaka [1,8] ✉

The liver stores glycogen and releases glucose into the blood upon increased energy demand. Group 2 innate lymphoid cells (ILC2) in adipose and pancreatic tissues are known for their involvement in glucose homeostasis, but the metabolic contribution of liver ILC2s has not been studied in detail. Here we show that liver ILC2s are directly involved in the regulation of blood glucose levels. Mechanistically, interleukin (IL)-33 treatment induces IL-13 production in liver ILC2s, while directly suppressing gluconeogenesis in a specific Hnf4a/G6pc-high primary hepatocyte cluster via Stat3. These hepatocytes significantly interact with liver ILC2s via IL-13/IL-13 receptor signaling. The results of transcriptional complex analysis and GATA3-ChIP-seq, ATAC-seq, and scRNA-seq trajectory analyses establish a positive regulatory role for the transcription factor GATA3 in IL-13 production by liver ILC2s, while AP-1 family members are shown to suppress IL-13 release. Thus, we identify a regulatory role and molecular mechanism by which liver ILC2s contribute to glucose homeostasis.

The risk factors for cardiovascular diseases and stroke are closely linked to pathological conditions that are collectively referred to as metabolic syndrome. Over time, excessive glucose levels cause organ complications in patients with diabetes, and fat accumulates in patients who are obese or overweight. The tissue deposition of lipids coupled with chronic inflammation causes decreased insulin sensitivity, termed insulin resistance, which can lead to hyperglycemia, hyperinsulinemia, and tissue damage, including damage to blood vessels. Hepatic gluconeogenesis is critical in the process of glucose homeostasis because it promotes the production of glucose from stored glycogen, amino acids, and lactate in response to hypoglycemia. Glucose-6-phosphatase (G6PC) and PCK1 are key enzymes in this process, and G6PC is especially indispensable because it converts glucose-6-phosphate into glucose in the final pathway step. Insulin suppresses excessive gluconeogenesis, but insulin resistance in the liver interferes with the proper regulation of glucose production. This means that lipid deposition in the liver under obese conditions weakens insulin signaling in hepatocytes, leading to excessive

[1]Department of Molecular Diagnosis, Graduate School of Medicine, Chiba University, Chiba, Japan. [2]Department of Endocrinology, Hematology and Gerontology, Graduate School of Medicine, Chiba University, Chiba, Japan. [3]Department of Immunology, Graduate School of Medicine, Chiba University, Chiba, Japan. [4]Department of Immunology, Tokai University School of Medicine, Isehara, Kanagawa, Japan. [5]Department of Systems Medicine, Graduate School of Medicine, Chiba University, Chiba, Japan. [6]Department of Medical Physiology, Chiba University, Graduate School of Medicine, Chiba, Japan. [7]Institute for Advanced Academic Research, Chiba University, Chiba, Japan. [8]Research Institute of Disaster Medicine, Chiba University, Chiba, Japan. [9]AMED-CREST, AMED, Otemachi, Tokyo, Japan. ✉e-mail: tomoaki@restaff.chiba-u.jp

gluconeogenesis and blood glucose elevation. Hepatic insulin resistance is a pathological hallmark of nonalcoholic fatty liver disease (NAFLD)[1–3], and its prevalence continues to increase. The drug metformin is administered to patients with diabetes to improve insulin resistance and mechanistically exerts its effects by suppressing hepatic gluconeogenesis. In addition to the current approaches, improvement of gluconeogenesis has attracted the attention of researchers as a medical treatment option.

Group 2 innate lymphoid cells (ILC2) are innate cells of lymphoid origin that develop from a common lymphoid progenitor and play pivotal roles in immune protection against helminth infection by secreting type 2 cytokines such as interleukin (IL)-4, IL-5, IL-9, and IL-13. Inflammation associated with type 2 immunity mediates responses to tissue repair, goblet cell hyperplasia, and autoimmune diseases such as arthritis. ILC2s are regulated by soluble factors, including cytokines and hormones, and IL-33 is known as a major activator of ILC2s. In addition to known roles in immunity[4–6], ILC2s have beneficial roles in metabolic disorders, and they reportedly limit adiposity by increasing caloric expenditure of beige adipocytes in white adipose tissue[7,8]. Another report has suggested that IL-33-activated islet ILC2s produce IL-13 and accelerate insulin secretion via the production of retinoic acid in myeloid cells[9]. Mallat and colleagues reported that ILC2-derived IL-13 protects against atherosclerosis in aortic tissues[10]. The functions of ILC2s in the liver were previously reported to be related to intrahepatic and extrahepatic fibrosis[11,12], and ILC2s in the liver appear to respond to physiological conditions. However, the metabolic functions of ILC2s in the liver and whether they differ from the functions of ILC2s found in adipose or pancreatic tissues remain unknown.

Here, we report that IL-33 treatment increases liver ILC2 numbers, decreases blood glucose levels, and downregulates the gluconeogenic enzyme *G6pc* in the livers of both normal and obese mice. However, these effects are absent in ILC2-null NSG mice, indicating that IL-33-mediated suppression of gluconeogenesis is dependent on ILC2s. The results of single-cell RNA sequencing (scRNA-seq) analysis of liver tissue show that liver ILC2s suppress gluconeogenesis in the *G6pc*-high hepatocyte cluster, possibly through downregulation of *G6pc* expression via signal transducer and activation of transcription 3 (STAT3) signaling. In addition, we identify AP-1 family as GATA3-binding proteins and that JunB suppresses GATA3 function, resulting in reduced IL-13 expression. These observations uncover the additional function of liver ILC2s and the role of the AP-1 family in GATA3 signaling for IL-13 regulation. Our findings provide insights into the potential functions of liver ILC2s in the modulation of gluconeogenesis and blood glucose levels and may present a new strategy for diabetes treatment.

## Results

### IL-33 lowers blood glucose levels by limiting gluconeogenesis via liver ILC2s

As increases in the levels of gluconeogenic enzymes are significant contributing factors to blood glucose elevation in obese humans and mice, we first investigated whether ILC2 activation could reduce blood glucose and gluconeogenic enzyme levels in the livers of obese mice. We intraperitoneally (i.p.) injected leptin receptor-deficient (db/db) obesity model mice with phosphate-buffered saline (PBS) or 0.5 μg of recombinant IL-33 (rIL-33) to activate ILC2s for 5 consecutive days. rIL-33 injection decreased fasting blood glucose levels and reduced the gene expression levels of liver *G6pc*, a rate-limiting enzyme of gluconeogenesis (Fig. 1a, b), which is consistent with a previous report[13]. T cells are major immune cells found in liver tissue that also respond to rIL-33; therefore, we compared the parameters of glucose metabolism among specific immune cell-deficient models to clarify which cell types contributed to the rIL-33-induced glucose-lowering effect. Nude (*Foxn*[nu/nu]) mice and NOD/Scid/Il2Rγ[null] (NSG) mice were used as deficiency models lacking mature T cells and both T and ILC2s cells, respectively. In wild-type (wt) and nude mice, rIL-33 injection

decreased fasting blood glucose levels (Fig. 1c), significantly reduced fasting insulin levels (Fig. 1d), and increased liver glycogen contents (Fig. 1e). However, these responses were not observed in NSG mice, indicating that the rIL-33-induced glucose-lowering effect was mediated by ILC2s in addition to glycogen storage in the liver.

We then subjected mice to a pyruvate tolerance test (PTT) to determine whether rIL-33 could suppress gluconeogenesis and glucose elevation in wt mice. As expected, rIL-33 injection suppressed blood glucose elevation during PTT in wt and nude mice but not in NSG mice (Fig. 1f and Supplementary Fig. 1a). Similar to the test with pyruvate, we also compared the effect of rIL-33 in a challenge test with glycerol, another substrate of gluconeogenesis, in rIL-33-responsive wt and nude mice. We confirmed that rIL-33 suppressed glucose elevation in both types of mice subjected to the glycerol tolerance test (GlycerolTT) (Supplementary Fig. 1b). For de novo glucose production, pyruvate required both G6PC and PCK1, whereas glycerol required G6PC but bypassed PCK1. These results suggested that the G6PC-dependent process of gluconeogenesis was one of the sites of action of rIL-33. Furthermore, in an insulin tolerance test (ITT), rIL-33 injection suppressed blood glucose recovery during the late phase (90 – 240 min) in wt mice but not in NSG mice (Fig. 1g), consistent with a previous report describing glucose recovery during the late phase of ITT as being dependent on gluconeogenesis[14]. We also examined the gene expression of gluconeogenic enzymes in the liver tissues of PBS- or IL-33-treated wt nude and NSG mice. Consistent with the in vivo results, the expression of several gluconeogenic enzymes (*G6pc*, *Pck1*, and *Hnf4a*) was significantly suppressed by IL-33 treatment in both wt and nude mice (Supplementary Fig. 1d). *Fbp1* and *Pcx* were inhibited somewhat, although not significantly, by IL-33 treatment in wt mice. We also found that none of the genes were significantly changed by IL-33 in NSG mice. These data suggested that the effect on gluconeogenic enzymes was mediated by ILC2s. Taken together, these findings indicated that rIL-33 administration suppressed gluconeogenesis via an ILC2-dependent mechanism.

To address whether ILC2s mediated the blood glucose-lowering effect of rIL-33, we intravenously transferred liver ILC2s (defined as Lin⁻Thy1⁺IL-7Rα⁺ST2⁺) into ILC2-deficient NSG mice and then measured the blood glucose levels of the mice after 5 days of daily rIL-33 or PBS administration (Fig. 1h). We confirmed the presence of ILC2s in the livers of ILC2-transferred NSG mice (Fig. 1i). While rIL-33 had no effect on fasting blood glucose levels in NSG mice that did not receive transfers (left two groups in Fig. 1j), rIL-33 reduced the fasting blood glucose levels of NSG mice that received ILC2 transfers (right group in Fig. 1j). Thus, the results of the transfer experiments indicated that rIL-33 reduced blood glucose levels via liver ILC2s. Collectively, these data showed that liver ILC2s limited gluconeogenesis in response to rIL-33 injection.

To explore whether liver ILC2s played roles in physiological models, we evaluated liver ILC2s in a 3-month normal diet (ND) or high-fat diet (HFD)-fed model. HFD-fed mice showed clear increases in fasting blood glucose, body weight, and blood glucose in the PTT and ITT (Supplementary Fig. 2a–c). These data confirmed that gluconeogenesis was enhanced in this model. We next performed flow cytometry analysis and found clear expansion of ILC2s in livers from the HFD-fed mice compared with those of the ND-fed mice (Supplementary Fig. 2d). These data might suggest that ILC2s play a compensatory role in this model.

### *Il13* is highly expressed in liver ILC2s, as revealed by scRNA-seq analysis

To characterize the molecular features of activated liver ILC2s following rIL-33 administration, we performed scRNA-seq analyses of sorted ILC2-enriched immune cell populations. We compared liver ILC2s and lung ILC2s to document the heterogeneity among organs, including differences in the presence and absence of rIL-33. Lung ILC2s

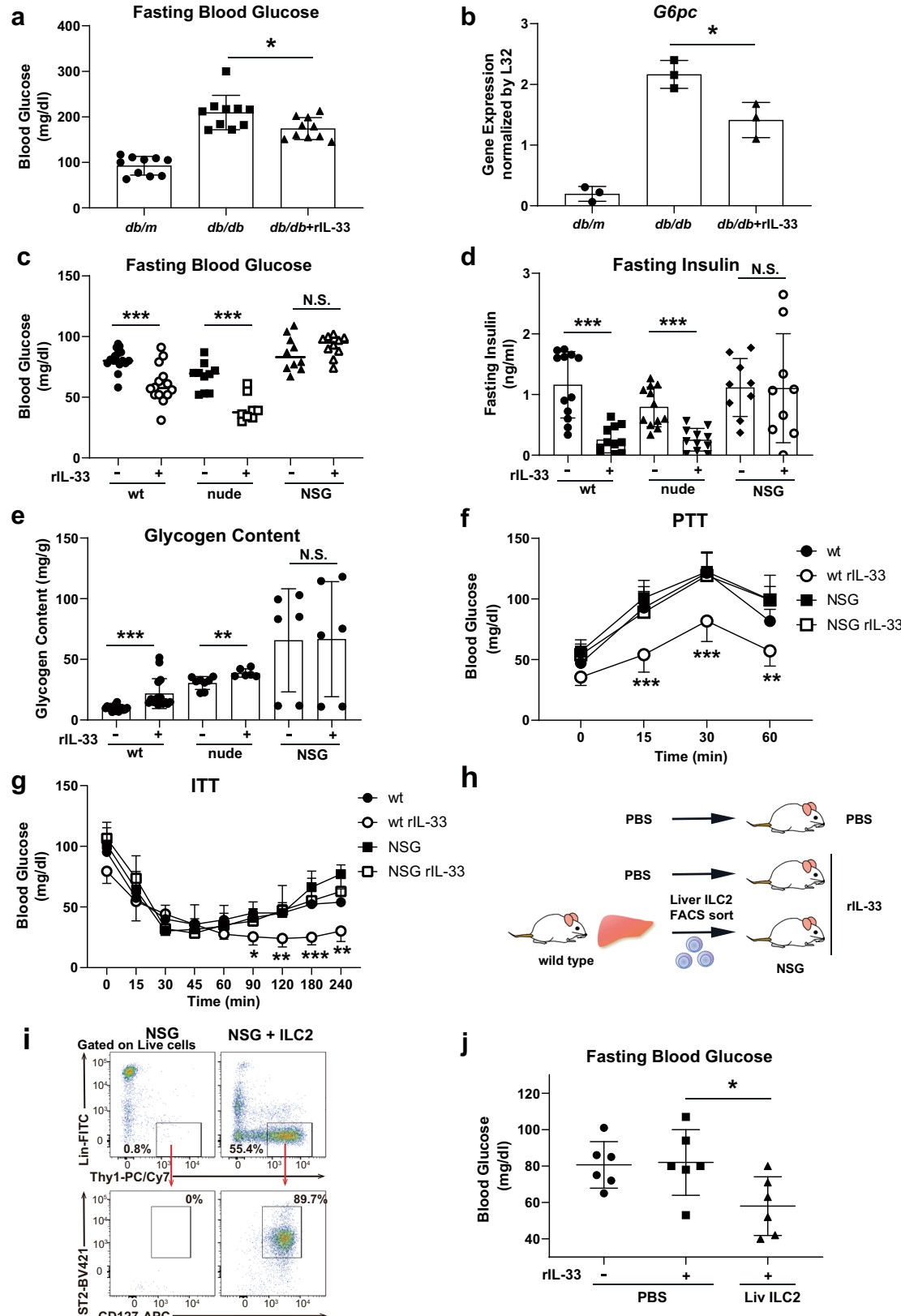

were used as well-characterized nonliver ILC2 controls. We examined the liver and lung immune cell types sorted from wt mice injected with PBS (control) or rIL-33 for 5 days. The cells were classified based on differentially expressed genes (DEGs) and the expression of known markers (Fig. 2a–c). Most of the sorted cells were defined as ILC2s, but other cell types were also identified according to marker profiles. We

excluded cells expressing non-ILC2 markers such as *Cd3*, *Cd11b*, *Fceri* and *Cd45r* and reclustered the remaining 14,026 ILC2s into 10 clusters (Fig. 3a). ILC2s from rIL-33-treated lungs were distributed to all clusters, whereas rIL-33-treated liver ILC2s were enriched in specific clusters (1, 2, 4, 6, and 8) (Fig. 3b, c). GATA3 is a crucial transcription factor involved in cell survival and cytokine production[15] in ILC2s, and Il-1

**Fig. 1 | Recombinant IL-33 (rIL-33) injection exerts a blood glucose-lowering effect by limiting gluconeogenesis, which may be mediated by liver group 2 innate lymphoid cells (ILC2). a** Fasting blood glucose levels in control (*db/m*) and *db/db* mice after 5 consecutive days of intraperitoneal phosphate-buffered saline (PBS) (*db/db*) or rIL-33 injections (*db/db* + rIL-33). The data are presented as the mean ± SD; *n* = 10 per group. **b** *G6pc* mRNA expression in liver tissues from control and *db/db* mice after 5 days of PBS or rIL-33 injections. The relative expression levels (normalized to L32) are shown with the SD (*n* = 3 per group). **c, d, e** Fasting blood glucose levels (**c**), fasting insulin levels (**d**), and liver glycogen contents (**e**) in wild-type (wt), nude and NOD/Scid/Il2Ry[null] (NSG) mice after 5 days of PBS or rIL-33 injections (wt rIL-33[-]: *n* = 16, wt rIL-33[+]: *n* = 14, nude rIL-33[-]: *n* = 10, nude rIL-33[+]: *n* = 8, NSG rIL-33[-]: *n* = 10, and NSG rIL-33[+]: *n* = 10 in c; wt rIL-33[-]: *n* = 12, wt rIL-33[+]: *n* = 11, nude rIL-33[-]: *n* = 12, nude rIL-33[+]: *n* = 11, NSG rIL-33[-]: *n* = 9, and NSG rIL-33[+]: *n* = 9 in d; wt rIL-33[+], wt rIL-33[+], nude rIL33[-], nude rIL33[+]: *n* = 8, NSG rIL-33[-], and NSG rIL-33[+]: *n* = 6 in e). **f, g** Blood glucose levels measured by the pyruvate tolerance test (wt, wt rIL-33, NSG: *n* = 7, NSG rIL-33: *n* = 6). **f, g** Blood glucose levels measured by the pyruvate tolerance test (wt, wt rIL-33, NSG: *n* = 7, NSG rIL-33: *n* = 6) (**f**) and insulin tolerance test (0.075 U/kg body weight insulin, wt: *n* = 8, wt rIL-33: *n* = 5, NSG: *n* = 7, NSG rIL-33: *n* = 5) (**g**) in wt and NSG mice were evaluated after 5 days of PBS or rIL-33 injections. **h** Schema of the transfer experiment. Briefly, NSG mice were intravenously injected with PBS or cultured liver ILC2s, and their fasting blood glucose levels were evaluated 5 days after injection with rIL-33. **i** Gating and frequency of Lin⁻Thy1⁺CD127⁺ST2⁺ ILC2s in the liver tissues of rIL-33-treated NSG mice with or without liver ILC2 transfer. A representative figure is shown (*n* = 5 per group). **j** Fasting blood glucose levels in NSG mice with or without liver ILC2 transfer (*n* = 6 per group). Unpaired one-sided Student's *t*-test. *P* < 0.05; **P* < 0.01; ***P* < 0.001. Each bar and its error bars represent the mean ± SD.

receptor-like 1 (Il1rl1) induces type 2 cytokine production in ILC2s[16]. Both *Gata3* expression and *Il1rl1* expression were detected in most clusters. Interestingly, the expression levels of these genes displayed unique patterns (Fig. 3d), indicating heterogeneous characteristics for ILC2s derived from different organs and treatment conditions. To characterize the unique features of each cluster, unbiased marker selection was performed (Fig. 3e), which identified GATA3-regulating genes (*Areg, Il1rl1* and *Gzma*). Therefore, we focused on the GATA3 downstream signaling pathway to identify whether effector molecules were specialized in each cluster. We identified significant differences in representative cytokines that characterized ILC2 function: high *Il13* expression in clusters 1, 2, 4, and 7; high *Il5* expression in clusters 7 and 9; high *Areg* expression in clusters 3, 7, and 8; and high *Arg1* expression in clusters 2, 6, and 9 (Fig. 2d). We confirmed that ILC2s expressed *Cd3g* or *Tcrg-C1* at very low levels compared with those in other cells, such as gamma delta T cells (Supplementary Fig. 3a). We also confirmed that the expression of TCRg in ILC2s was undetectable by flow cytometry analysis (Supplementary Fig. 3b).

Furthermore, we compared the overall gene expression levels across different treatment conditions in the liver and between liver and lung ILC2s. In addition to the gene profiles that were altered by rIL-33 stimulation (Fig. 3f, left), liver ILC2s presented a distinct gene profile that differed from that of lung ILC2s (Fig. 3f, right). In the context of GATA-dependent genes, *Il13* expression was significantly upregulated in liver ILC2s compared with lung ILC2s (Fig. 3f, right), but this pattern was not observed for other genes downstream of GATA3, such as *Il1rl1, Il5*, and *Areg* (Fig. 3g), and *Gata3* expression did not correlate with *Il13* expression (Fig. 2e). We confirmed that IL-13 expression was significantly increased in liver ILC2s compared to lung ILC2s using flow cytometry (Fig. 3h, i). These data suggested that the regulation of GATA3-dependent cytokines differed under different conditions and across distinct organs even though GATA3 was a key transcription factor in ILC2s. In particular, the expression of IL-13 was more strongly induced in liver ILC2s than in lung ILC2s in response to IL-33.

## Liver ILC2-derived IL-13 is involved in the blood glucose-lowering effect of IL-33

The IL-13 and STAT3 signaling pathway has been reported to directly suppress hepatic glucose production[17]; we therefore hypothesized that ILC2-derived IL-13 mediates a suppressive effect on gluconeogenesis. First, we investigated whether IL-13 limited gluconeogenesis in wt mice. The PTT after injection of either PBS or 1 μg of recombinant IL-13 (rIL-13) confirmed that rIL-13 injection significantly suppressed blood glucose elevation in response to pyruvate (Fig. 4a). As shown in Fig. 1c, NSG mice that lacked ILC2s did not exhibit an IL-33-induced glucose-lowering effect. To determine the role of IL-13 in this process, we measured IL-13 levels in specific immune cell-deficient models (nude and NSG mice) after rIL-33 treatment. The rIL-33 injection increased IL-13 levels in both liver tissues and blood serum of wt and nude mice. In

contrast, IL-13 levels remained unchanged in NSG mice (Fig. 4b, c). Next, we analyzed glucose metabolism following IL-33 stimulation in *Il13*-knockout mice. Although rIL-33 injection decreased fasting blood glucose levels and suppressed gluconeogenesis in *Il13*⁺/⁺ or *Il13*⁺/⁻ mice, *Il13*⁻/⁻ mice did not exhibit either glucose-lowering effects or gluconeogenesis suppression following rIL-33 injection (Fig. 4d, e). Finally, we transferred liver ILC2s derived from *Il13*⁺/⁺, *Il13*⁺/⁻ or *Il13*⁻/⁻ mice into ILC2-deficient NSG mice to confirm the contributions of the IL-33-IL-13 axis to the effects observed following rIL-33 injection (Fig. 4f). Transferring *Il13*⁺/⁻ ILC2s into NSG mice decreased fasting blood glucose levels in response to rIL-33 injection, but *Il13*⁻/⁻ ILC2s did not demonstrate significant glucose-lowering effects (Fig. 4g). Collectively, these data suggested that ILC2-derived IL-13 mediated the blood glucose-lowering effects of rIL-33. In particular, the glucose-lowering effects mediated by ILC2s were able to influence gluconeogenesis in hepatocytes. We also examined whether ILC2s played a similar role in a 3-month HFD-fed model. In 3-month HFD-fed mice, the numbers of IL-13(+) ILC2s clearly expanded in liver tissue (Supplementary Fig. 2e, f). Il13 expression in the whole liver was also increased, as confirmed by real-time quantitative PCR (RT–qPCR) (Supplementary Fig. 2g). In addition, neutralization of IL-13 significantly suppressed blood glucose elevation in the PTT in mice fed a HFD for 3 months (Supplementary Fig. 2h). Collectively, these data suggested that IL-13 suppressed gluconeogenesis in the HFD-fed obesity model.

To examine the direct effect of hepatic ILC2s on the gluconeogenesis of hepatocytes, we cocultured hepatic ILC2s and primary hepatocytes (Fig. 4h). To confirm the effect of ILC2-derived IL-13, we also tested the effect of additional IL-13 neutralization. As expected, several gluconeogenic enzymes (*G6pc, Pck1, Hnf4a, Fbp1, Pcx* and *Ppargc1a*) were downregulated by coculture with ILC2s, and most of the effect was abolished by IL-13 neutralization (Fig. 4i). These data support the concept that ILC2s directly suppress gluconeogenesis in hepatocytes via IL-13. Furthermore, to evaluate the effect of ILC2s on metabolites in hepatocytes, we next cocultured ILC2s and hepatocytes and then assessed the metabolome in hepatocytes (Fig. 4h). Coculture of ILC2s significantly altered the amino acids in hepatocytes, especially those in the pyruvic acid pathway: Ala, Cys and Gly (Fig. 4j). These three amino acids are classified as glucogenic amino acids, which can be substrates for gluconeogenesis. Additionally, we examined the metabolome in the gluconeogenesis-related pathway (Fig. 4k). Coculture with ILC2s significantly increased fructose-1,6-bisphosphate (F1,6P) and pyruvic acid levels and decreased 3-phosphoglyceric acid (3-PG), 2-phosphoglyceric acid (2-PG), and phosphoenolpyruvic acid (PEP) levels. The increases in F1,6P and pyruvic acid levels may have been related to decreased expression of *Fbp1* and *Pcx* (Fig. 4i, k). In addition, the decrease in PEP might have been partly attributable to decreased expression of *Pck1*. With regard to the tricarboxylic acid (TCA) cycle, intermediate metabolites tended to accumulate after coculture with ILC2s, and citric acid and malic acid levels were especially significantly increased (Fig. 4l). These results suggested that liver ILC2-derived IL-13

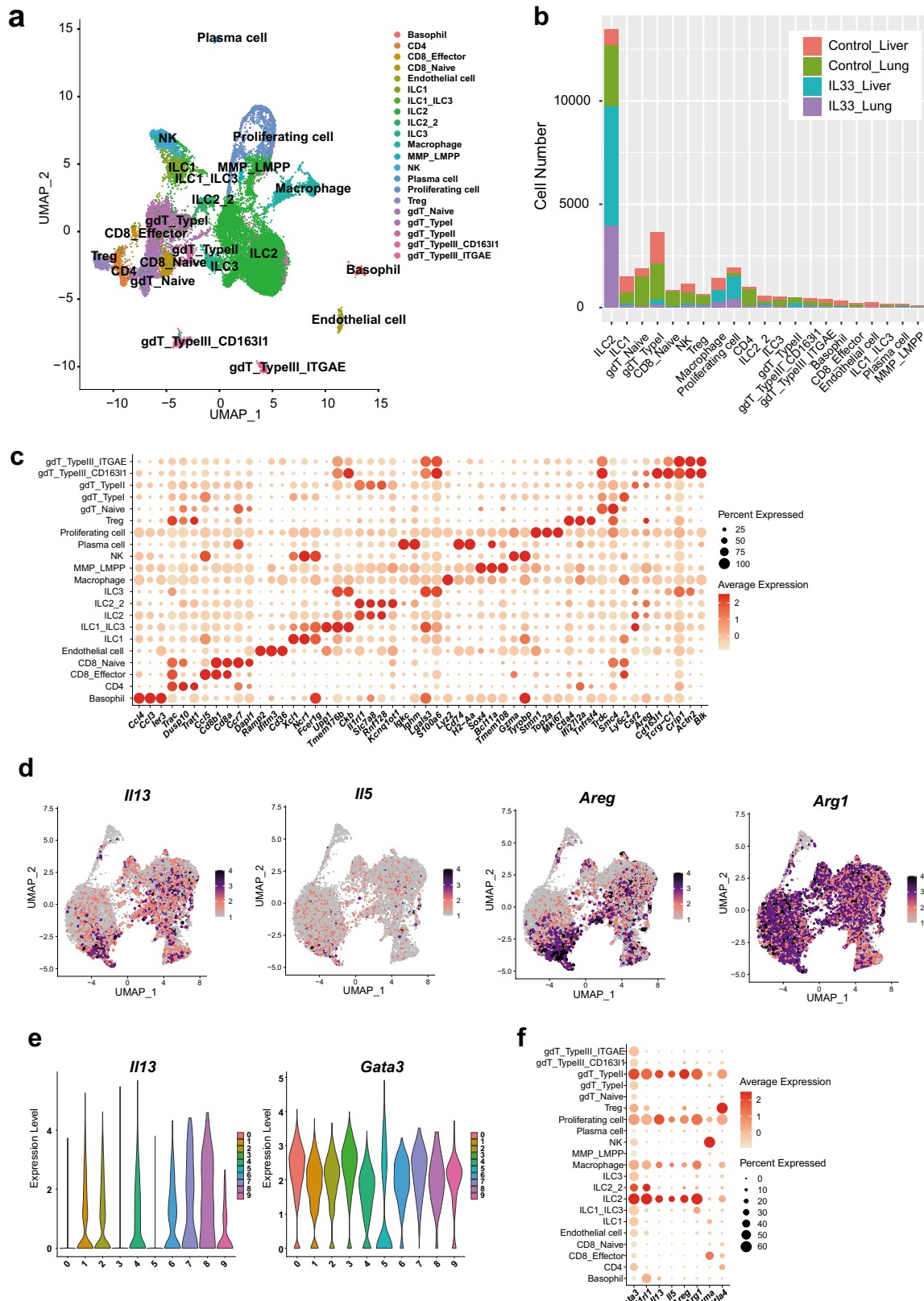

**Fig. 2 | scRNA-seq reveals that liver group 2 innate lymphoid cells (ILC2) highly express *Il13*, which may contribute to the blood glucose-lowering effect of IL-33. a** scRNA-seq data (*n* = 31,186 single immune cells) for phosphate-buffered saline- and recombinant IL-33 (rIL-33)-treated liver or lung ILC2s, shown as nonlinear representations of the top 50 principal components; the cells are colored based on cell type. **b** Cell numbers of the clusters within each group, as defined by the treatment condition and tissue. **c** Differentially expressed genes in each cell type, as defined by the FindAllMarkers function. **d** Uniform manifold approximation and projection (UMAP) plots of all immune cells showing the expression of Gata3 downstream genes (*Il13*, *Il5*, *Areg*, and *Arg1*). **e** Violin plot showing the expression of *Il13* and *Gata3* in each cluster. **f** Dot plot showing the expression of Gata3 downstream genes in each cell type.

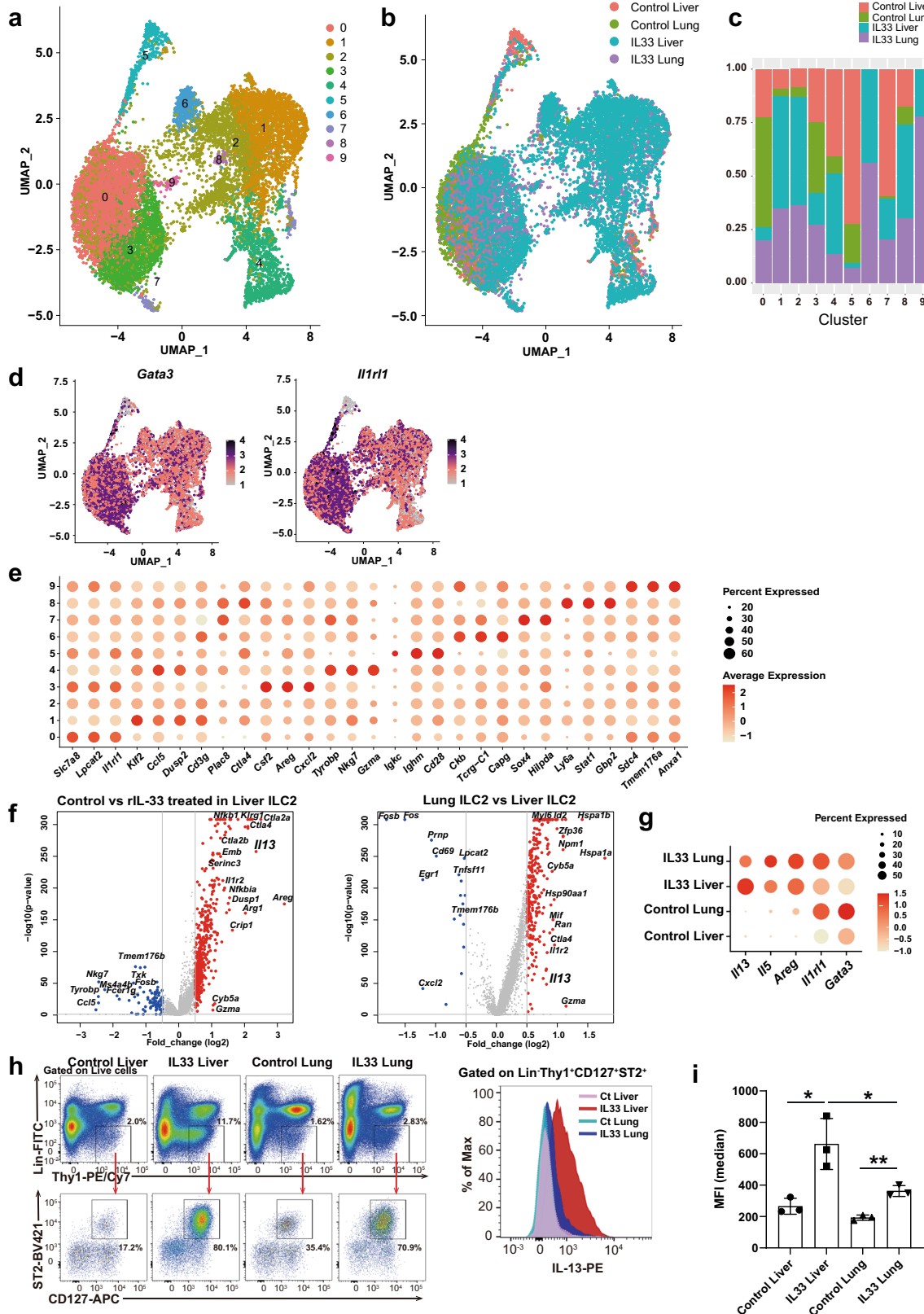

directly suppressed the gene expression of gluconeogenic enzymes and significantly altered the metabolite content in hepatocytes.

To further investigate the effects of IL-13 on hepatocytes, we also performed RNA-seq analyses on primary hepatocytes treated with rIL-13. Primary hepatocytes were incubated with rIL-13 for 2 h and then stimulated with cAMP/glucagon because glucagon promotes

gluconeogenesis through the cAMP signaling pathway (Supplementary Fig. 4a). To identify gluconeogenesis-related genes affected by rIL-13, DEGs were measured after cAMP stimulation and rIL-13 treatment. Pathway analysis showed that glucose metabolism was affected by the presence of rIL-13 (Supplementary Fig. 4b). In particular, forkhead box O (FOXO) signaling is known to induce liver gluconeogenesis by

**Fig. 3 | scRNA-seq reveals that liver group 2 innate lymphoid cells (ILC2) highly express Il13, which may contribute to the blood glucose-lowering effect of IL-33. a–i** Liver and lung ILC2s were sorted from wild-type BALB/c mice intraperitoneally (i.p.) injected with phosphate-buffered saline (PBS) (control) or recombinant IL-33 (rIL-33) for 5 days. ILC2s from each of the four groups were profiled by droplet-based scRNA-seq. **a, b** The scRNA-seq data (*n* = 14,026 single ILC2s) across all four groups of ILC2s are shown as nonlinear representations of the top 50 principal components; the cells are colored according to uniform manifold approximation and projection (UMAP)-based clusters (**a**) or according to treatment and tissue type (**b**). Subclustering of ILC2s was performed with a resolution of 0.5. **c** Proportions of the clusters within each group as defined by treatment condition

and tissue source. **d** UMAP plots showing the expression of the innate lymphoid cell (ILC) markers *Gata3* and *Il1rl1* in all ILC2 samples. **e** Dot plot showing the differentially expressed genes (DEG) in each cluster as defined by the FindAllMarkers function. **f** Volcano plots showing the DEGs between PBS-treated liver and rIL-33-treated liver samples (left) or between rIL-33-treated liver and rIL-33-treated lung samples (right). **g** Expression levels of representative Gata3 downstream genes, as grouped by treatment and tissue. **h** Gating for Lin⁻Thy1⁺CD127⁺ST2⁺ ILC2s (left) and frequencies and fluorescence intensities of IL-13 in liver and lung ILC2s from PBS- or rIL-33-treated mice (right) (*n* = 3 per group). **i** Mean fluorescence intensity (MFI) of IL-13 in ILC2s in each group (mean ± SD, *n* = 3). Unpaired one-sided Student's *t*-test. *$P < 0.05$; **$P < 0.01$; ***$P < 0.001$.

activating genes such as *G6pc* and *Pck1* and was mostly downregulated by rIL-13 treatment in the presence of cAMP or glucagon (Supplementary Fig. 4c). In addition, we confirmed the gene expression of *G6pc* and *Pck1* by RT–qPCR analyses (Supplementary Fig. 4d). Similar to the RNA-seq results, the RT–qPCR analysis results showed that these genes were upregulated by cAMP or glucagon treatment and suppressed by rIL-13 treatment, indicating that IL-13 directly suppressed gluconeogenesis in hepatocytes. Thus, these results suggested that liver ILC2-derived IL-13 inhibited gluconeogenesis in hepatocytes.

### The gluconeogenesis pathway is suppressed in certain hepatocyte clusters via the IL-33-IL-13 axis in vivo, as indicated by scRNA-seq with cell–cell interaction analysis

To clarify the underlying mechanism of action through which the IL-33-IL-13 axis affected hepatocytes, we performed scRNA-seq analyses of primary hepatocytes from fasted mice treated with rIL-33 or PBS for 5 days (Fig. 5a). A total of 11,327 cells were individually analyzed, and the cell types were classified by markers (Fig. 5b). A total of 10,186 cells were defined as hepatocytes (e.g., expressing *Alb*, *Pck1*, *G6pc*, and *Hnf4a*), which were categorized into five groups based on their heterogeneity (Fig. 5b). Cells from both the PBS-treated and rIL-33-treated groups were distributed in all five clusters, but the frequencies were somewhat altered by IL-33 stimulation (Supplementary Fig. 5a, b). The graph of the cell cycle phase also shows that there were some differences among the hepatocyte clusters (Supplementary Fig. 5c). Next, to explore the traits of each hepatocyte cluster, we observed the representative zonation markers and DEGs in each cluster. Glul is highly expressed near the central vein (CV), and *Cdh1* is highly expressed in the portal node (PN)[18]. We found that *Glul* expression was higher in the Hepatocyte1 and Hepatocyte2 clusters than in the other clusters (Fig. 5c left panel). In contrast, *Cdh1* expression was higher in the Hepatocyte4 and Hepatocyte5 clusters. These data suggest that the hepatocyte cluster reflects the location in which the hepatocytes reside. DEG analysis for each hepatocyte cluster revealed high expression of gluconeogenic enzymes, including *G6pc* and *Pck1*, especially in the Hepatocyte4 and Hepatocyte5 clusters (Fig. 5c right panel, Supplementary Fig. 5d). In addition, Gene Ontology (GO) analysis of the DEGs in each cluster suggested that Hepatocyte4 genes played roles in metabolic processes (Supplementary Fig. 5e).

We next investigated which hepatocyte cluster interacts with ILC2s. To answer this question, we combined hepatocytes and ILC2s, as shown in Fig. 3b, and performed interaction analysis. The liver ILC2-enriched clusters (1, 2, 4, 6, and 8) displayed increased interactions with the Hepatocyte4 cluster, followed by the Hepatocyte5 cluster (Fig. 5d and Supplementary Fig. 5f). Moreover, the interaction between IL-13 and IL-13R was strongly observed in the Hepatocyte4 cluster (Fig. 5e). We also performed immunofluorescence and found that some ILC2s existed near *Pck1*-high hepatocytes (Fig. 5f). To predict the regulatory factors affected by IL-33 stimulation, we performed an upstream gene analysis of all identified DEGs between PBS-treated and rIL-33-treated hepatocytes. The most enriched regulator was *Hnf4a* in this analysis, and the expression of genes downstream of *Hnf4a* was significantly decreased in the IL-33-treated cells (Fig. 5g, h). *Hnf4a* is

known to play roles in glucose metabolic processes, including gluconeogenesis, in the liver. In fact, several reports, including one on the results of an HNF4A chromatin IP (ChIP) assay, have shown that HNF4A binds to the promoter regions of *G6pc* and *Pck1*, leading to enhanced expression[19]. The enrichment of *G6pc* and *Pck1* expression was confirmed by uniform manifold approximation and projection (UMAP, Fig. 5i upper panel), and rIL-33 injection significantly downregulated *G6pc* expression, especially in the Hepatocyte4 cluster (Fig. 5i lower panel). In support of this finding, analysis of the overall gene distribution revealed that rIL-33 treatment significantly downregulated *G6pc* and *Pck1* levels (Fig. 5j).

A previous report has suggested that the IL-13-STAT3 pathway promotes the downregulation of gluconeogenic genes. IL-13 binds to the IL-13 receptor α1 subunit (encoded by *Il13ra1*) in a receptor complex expressed on hepatocytes that also contains IL-4Rα (encoded by *Il4ra*). We confirmed that the population of clusters that interacted with ILC2s (Hepatocyte4 and Hepatocyte5) highly expressed genes associated with the IL-13-STAT3 pathway, including *Il3ra1*, *Il4ra*, *Stat3*, and *Stat6* (Fig. 5k). We also confirmed direct binding of STAT3 upstream of the *G6pc* and *Pck1* transcription start sites (TSSs) using a ChIP assay method reported by Lee et al. (Fig. 5l). To confirm that gluconeogenic enzymes were suppressed by IL-13/STAT signaling, we performed a knockdown experiment. In the presence of IL-13, suppression of *G6pc* or *Pck1* expression on hepatocytes was significantly blocked by knockdown of *Il13ra1*, *Il4ra*, and *Stat3* but not by knockdown of *Stat6* (Fig. 5m). These results suggest that IL-13 directly suppresses gluconeogenic enzymes on hepatocytes via IL-13 receptor/Stat3 signaling. We also found that *Hnf4a* signaling was a common enriched pathway in both DEGs between hepatocyte clusters and DEGs between the PBS- and IL-33-treated groups (Fig. 6a). In addition, *Hnf4a* expression itself was elevated in Hepatocyte4 cells but was significantly suppressed by IL-33 treatment (Fig. 6b). Therefore, we hypothesized that the downregulation of *Hnf4a* might be mediated by *Stat3* activation. To confirm this possibility, we searched the STAT3 binding motif using the JASPAR database and found a STAT3 binding motif approximately 1.5 kb upstream from the *Hnf4a* TSS (Fig. 6c). In addition, we found that rIL-13 treatment promoted STAT3 binding upstream of *Hnf4a* in primary hepatocytes (Fig. 6d). As the hepatocyte isolation method includes a digestion process, we verified that at least most hepatocytes in the UMAP analysis were viable cells. Notably, cell death markers (*Susd6*, *Fas*, *Bax*, *Bcl2*) were rarely expressed on hepatocytes (Fig. 6e). In addition, the GO enrichment list for DEGs in Hepatocyte4 scarcely included terms related to cell death or stress; most GOs were related to metabolic processes (e.g., steroid metabolic process, monocarboxylic acid metabolic process) (Fig. 6f).

### The Gata3-mediated complex involves AP-1 family members to achieve transcriptional regulation in ILC2s

GATA3 serves as a central mediator of cytokine expression associated with type 2 immunity in ILC2s and T helper (Th)2 cells. However, *Il13* expression levels in ILC2s were not uniformly altered in response to IL-33 stimulation despite GATA3 being activated to a similar extent (Figs. 2d and 3d). This finding indicated that the transregulation of *Il13*

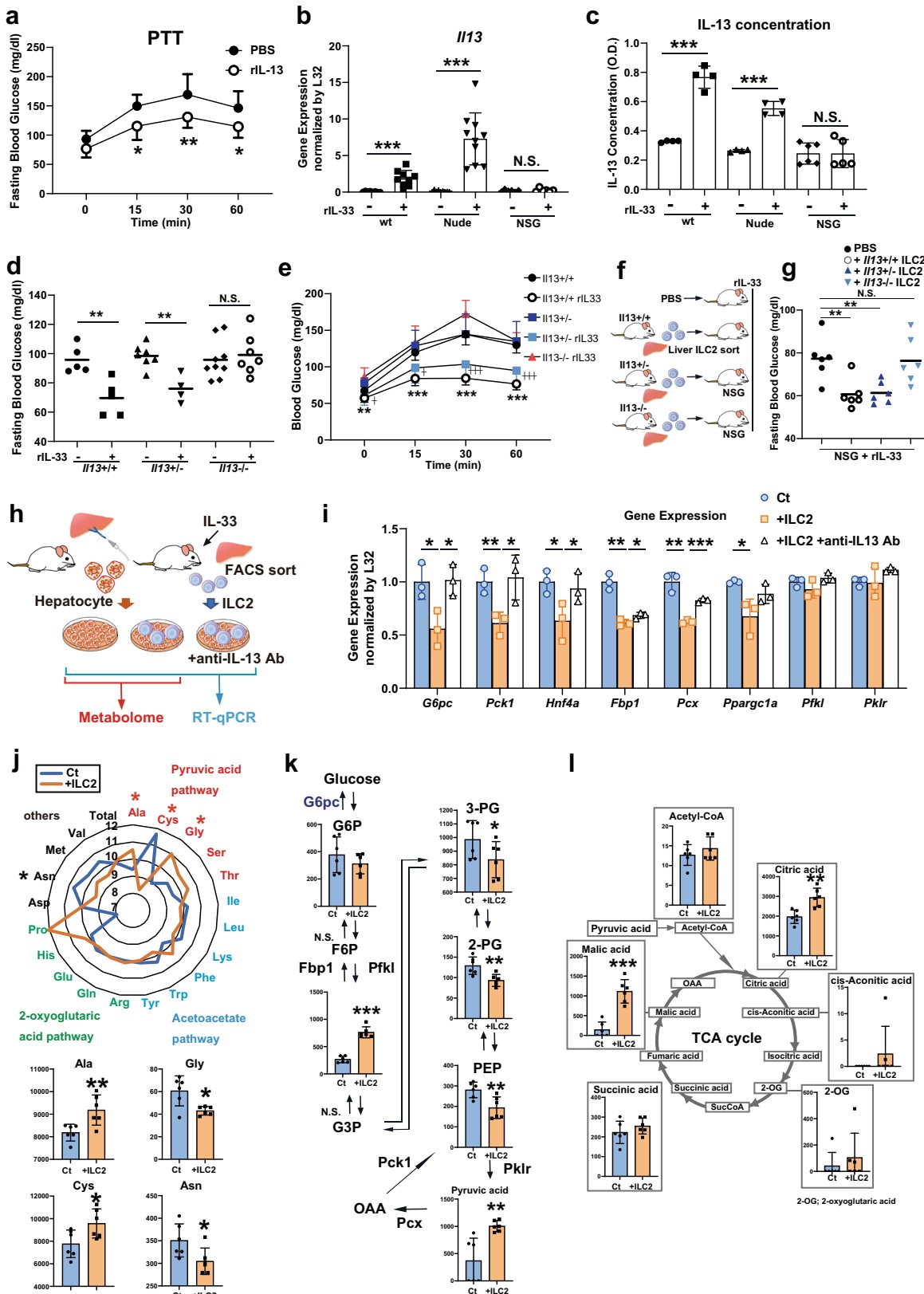

did not simply depend on the presence of GATA3. We previously showed that chromodomain helicase DNA-binding protein 4 (Chd4) plays a role in *Il13* induction in Th2 cells as a component of the GATA3 functional complex[20,21]. Therefore, we focused on GATA3-interacting proteins that are functionally specific in ILC2s. To identify potential GATA3-interacting proteins under conditions of IL-33 stimulation in

ILC2s, we used GATA3 IP and subsequent LC–MS/MS (IP-LC–MS/MS) analysis[20,21] of liver and lung ILC2s (Fig. 7a, upper panel). In total, 459 and 352 GATA3-binding proteins were detected in liver and lung ILC2s, respectively, which were then categorized into nuclear proteins and epigenetic factors (Fig. 7a, lower panel). We selected 26 and 20 proteins from liver and lung ILC2s as transcriptional regulators and GATA3

**Fig. 4 | Liver group 2 innate lymphoid cell (ILC2)-derived IL-13 directly suppresses gluconeogenesis in hepatocytes. a** Blood glucose levels in wild-type (wt) mice treated with phosphate-buffered saline (PBS) or IL-13 as measured by the pyruvate tolerance test (PTT) (n = 8 per group). **b, c** *Il13* expression levels in liver tissue (normalized to *L32*) (**b**) and serum IL-13 concentrations (**c**) in wt, nude, and NOD/Scid/Il2Ry[null] (NSG) mice intraperitoneally (i.p.) injected with PBS or recombinant IL-33 (rIL-33) for 5 consecutive days (wt rIL-33[-]: n = 8, wt rIL-33[+]: n = 9, nude rIL33[-]: n = 9, nude rIL33[+]: n = 10, NSG rIL-33[-] n = 6, and NSG rIL-33[+]: n = 6 in b; wt rIL-33[-]: n = 4, wt rIL-33[+]: n = 4, nude rIL33[-]: n = 4, nude rIL33[+]: n = 4, NSG rIL-33[-] n = 6, and NSG rIL-33[+]: n = 5 in c). **d, e** Fasting blood glucose (**d**) and blood glucose levels as measured by PTT (**e**) in *Il13*[+/-], *Il13*[+/-] and *Il13*[-/-] mice after 5 days of rIL-33 injection (*Il13*[+/+]rIL-33[-]: n = 5, *Il13*[+/+]rIL-33[+]: n = 5, *Il13*[+/-]rIL-33[-]: n = 7, *Il13*[+/-]rIL-33[+]: n = 4, *Il13*[-/-] rIL-33[-]: n = 9 and *Il13*[-/-] rIL-33[+]: n = 7 in (**d**), *Il13*[+/+]rIL-33[-]: n = 7, *Il13*[+/+]rIL-33[+]: n = 5, *Il13*[+/-]: n = 4, *Il13*[+/-]rIL-33: n = 7 and *Il13*[+/-]rIL-33: n = 6 in **e**). **f** Schema of the experiment. PBS, liver ILC2s from *Il13*[+/+] mice or liver ILC2s from *Il13*[-/-] mice were injected intravenously into NSG mice. rIL-33 was i.p. injected for 5 consecutive days, and fasting blood glucose levels were measured after 3 and 5 days. **g** Fasting blood glucose levels in NSG mice without ILC2 transfer

and with *Il-13*[+/-], *Il-13*[+/-] or *Il-13*[-/-] liver ILC2 transfer at days 3 and 5 (n = 3). **h** Experimental schema for coculture and subsequent real-time quantitative PCR (RT–qPCR) and metabolome analysis. After 6 h of coculture with ILC2s, the hepatocytes were used for RT–qPCR and metabolomic analysis. **i** RT–qPCR analysis of gluconeogenesis-related genes in primary hepatocytes cocultured with ILC2s (Ct: control hepatocytes, +ILC2: hepatocytes cocultured with ILC2s, + ILC2 + anti-IL13 Ab: hepatocytes cocultured with ILC2s in the presence of an anti-IL-13 antibody; n = 3 per group). **j, k, l** Metabolomic analysis of primary hepatocytes after coculture with ILC2s (Ct: control hepatocytes, +ILC2s: hepatocytes cocultured with ILC2s; n = 6 per group). **j** Amino acid metabolites categorized into the pyruvic acid pathway, acetoacetate pathway, 2-oxoglutaric acid pathway and others. **k** Metabolites in the gluconeogenesis-related pathway. (G6P; glucose 6-phosphate, F6P; fructose 6-phosphate, F1,6P; fructose 1,6-diphosphate, G3P; glyceraldehyde 3-phosphate, 3-PG; 3-phosphoglyceric acid, PEP; phosphoenolpyruvic acid, OAA; oxaloacetate, Pklr; pyruvate Kinase L/R, Pcx; pyruvate carboxylase). **l** Metabolites in the tricarboxylic acid (TCA) cycle. Unpaired one-sided Student's *t*-test. $^{*}P < 0.05$; $^{**}P < 0.01$; $^{***}P < 0.001$. Each bar and its error bars represent the mean ± SD.

---

complex components, respectively (Fig. 7b). The top-ranked proteins were similar between liver ILC2s and lung ILC2s, especially members of the activating protein 1 (AP-1) family of transcription factors, such as JunB and JunD.

Next, we performed GATA3 ChIP sequencing (ChIP-seq) and assay for transposase-accessible chromatin with high-throughput sequencing (ATAC-seq) to determine whether the active transcriptional targets of GATA3 were altered in liver ILC2s and lung ILC2s. IL-33-stimulated open chromatin loci were defined by comparison between PBS- and rIL-33-treated cells in ATAC-seq. Consequently, we identified 7,031 differentially accessible regions (DAR), 1371 of which were bound by GATA3 in rIL-33-treated ILC2s in the liver (Fig. 7c and Supplementary Fig. 6c–e). Indeed, the upstream region of *Il13* became more open in rIL-33-treated ILC2s than in PBS-treated ILC2s (Fig. 7d). To predict the interacting partners of the GATA3 transcriptional complex, we searched for enriched enhancer motifs among the active GATA3-binding targets. A motif analysis revealed that AP-1, Etv2, and Runx1 binding motifs were enriched at these GATA3-bound DARs in liver ILC2s (Fig. 7c, right). GATA3 actively bound the *Il-13* genomic locus under IL-33 stimulation conditions in liver ILC2s, and the *Il13* enhancer locus included the motif for AP-1 (Fig. 7d). In particular, the JunB (also known as AP-1)-binding motif was significantly enriched in both cell types, suggesting that AP-1 family transcription factors, including JunB and JunD, bound GATA3 in ILC2s (Supplementary Fig. 6c). Based on this possibility, we searched for JunB binding motifs using the JASPAR database and found the motif in GATA3-binding peaks located 40 bases upstream (motif1) and approximately 2.9 kb upstream (motif2) from the *Il13* TSS (Fig. 7d). ChIP analysis confirmed that JunB bound to this region (Fig. 7e). Furthermore, to confirm the direct binding of GATA3-JunB on ILC2s, we performed IP western blotting. The ILC2 cell line ILC2/b6 was infected with a retroviral vector overexpressing FLAG-GATA3 or JunB. FLAG-IP and subsequent JunB western blotting clearly showed the binding of GATA3-JunB (Fig. 7f and Supplementary Fig. 7).

To further investigate the functional roles of GATA3-interacting molecules, we knocked down potential regulators using small interfering RNA (siRNA). The target molecules were selected from the liver ILC2 data according to the Mascot score derived from the GATA3-IP-LC−MS/MS results and the expression levels obtained from scRNA-seq (Fig. 7g). As Runx1 and Runx3 are known to function as crucial transcriptional regulators of IL-13 on ILC2s, we used si*Runx1* and si*Runx3* as experimental controls along with si*Gata3* (Fig. 7h). Sorted liver ILC2s were cultured with rIL-33 supplementation for 4 days, and siRNA oligonucleotides (targeting *Gata3*, *Junb*, *Jund*, *Pa2g4*, *Calr*, *Irf8*, *Runx1*, and *Runx3*) were transfected using electrophoresis. The cells were harvested 24 h later, and *Il13* mRNA levels were analyzed by RT−qPCR (Fig. 7i). As expected, *Il13* mRNA levels were suppressed by the

knockdown of *Gata3*, *Runx1*, or *Runx3*. On the other hand, siRNA targeting *Junb* significantly upregulated *Il13* mRNA levels, suggesting that AP-1 family members, including Junb, had a suppressive effect on *Il13* transcription. We also performed RNA-seq analyses using the same sample groups analyzed by RT−qPCR (Fig. 7h). The data were classified based on the responses to si*Gata3*, si*Junb*, and si*Runx1*, so we focused on the cluster that was downregulated by si*Gata3* and upregulated by si*Junb* (Fig. 7j). Taken together, the data suggested that AP-1 family members were involved in the GATA3 complex that formed in both liver and lung ILC2s. Moreover, AP-1 family members may have played suppressive roles in GATA3-dependent cytokine production in ILC2s.

## Different expression levels of GATA3-interacting AP-1 family members differentially activate ILC2s in an organ-specific manner

*Il13* induction in liver ILC2s was higher than that in lung ILC2s under IL-33 stimulation conditions; therefore, we investigated whether the suppressive effects against GATA3 activity mediated by AP-1 family members were associated with differential levels of *Il13* expression. First, we measured the expression levels of AP-1 family members in ILC2s using scRNA-seq data. Most of the genes in the AP-1 family were suppressed by IL-33 stimulation, and their expression levels in liver ILC2s were significantly lower than those in lung ILC2s (Fig. 8a). It has been reported that AP-1 family genes are rapidly upregulated with digestion stress; however, there was a clear difference between the control and IL-33-treated liver ILC2s even though the samples were digested with the same protocol and under the same conditions[22]. To confirm the function of the AP-1 family directly, we knocked down other AP-1 family members (*Jun*, *Fos*, *Fosb*, and *Fosl2*) and performed RT−qPCR for *Il13*. The results suggested that *Fos* and *Fosb* suppress *Il13* expression, while *Fosl2* increases it, on liver ILC2s (Fig. 8b). We also tested the effect of JunB overexpression on ILC2s and found that JunB overexpression suppressed *Il13* expression in both primary liver ILC2s and the cell line ILC2s (Fig. 8c).

Pseudotime analysis was used to clarify whether changes in *AP-1* family gene expression correlated with changes in *Il13* expression in liver ILC2s. The cells obtained from PBS-treated ILC2s were defined as the roots, and the response of the cells to IL-33 stimulation was analyzed. To characterize the end cluster containing activated liver ILC2s (cluster 3), we grouped similarly regulated genes into modules and then calculated the correlation between gene modules and clusters. This analysis showed that the end cluster containing activated ILC2s (cluster 3) was positively correlated with one specific module (Liver Module 6, Fig. 8d). Interestingly, the top GO term of the Liver Module 6 was TNF signaling, which was associated with *Il13* and *AP-1* family member-encoding genes. In contrast, data obtained from lung ILC2s

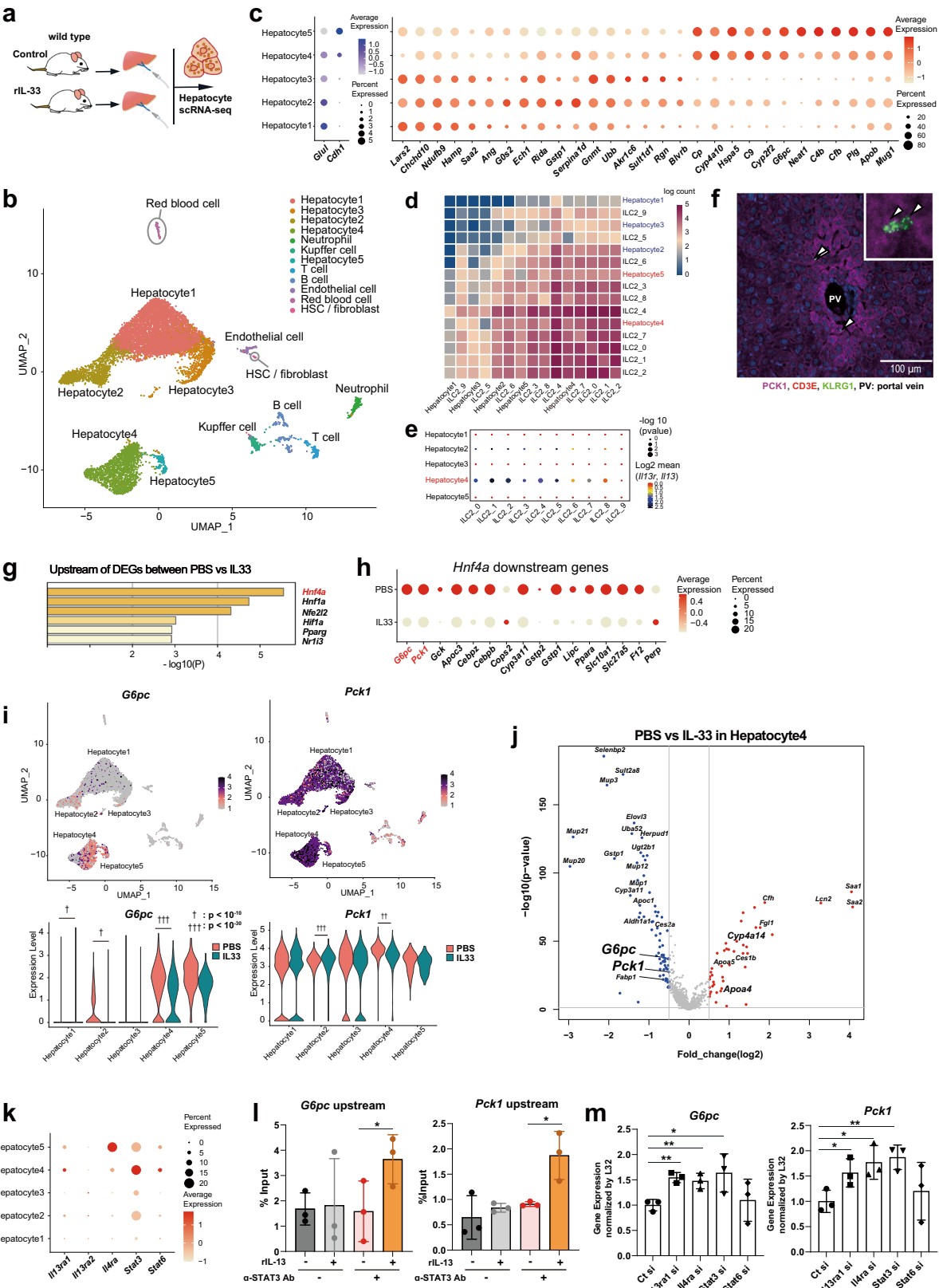

showed that the *AP-1* genes were not contained in the module (Lung Module 7) that was highly correlated with activated ILC2s (Fig. 8e). In Fig. 8f, the matrix shows the correlations of average gene expression in each cluster, revealing significant negative correlations between the expression of cytokines downstream of GATA3, such as *Il13*, and *AP-1* gene expression. However, these correlations were weaker in lung

ILC2s than in liver ILC2s (Fig. 8f). In detail, the course of *Il13* gene activation from the pseudotime analysis of liver ILC2s was negatively associated with *AP-1* gene activation, especially *Fos* and *Fosb* activation. However, this relationship was not observed in lung ILC2s (Fig. 8g). Taken together, the findings indicated that the expression of *AP-1* genes was suppressed during the course of ILC2 activation in the liver

**Fig. 5 | IL-33 suppresses gluconeogenesis in hepatocytes, especially in the cluster that most significantly interacts with liver group 2 innate lymphoid cells (ILC2). a** Schema of the experiment. Hepatocytes were isolated from phosphate-buffered saline (PBS)- and recombinant IL-33 (rIL-33)-injected mice, and then subjected to scRNA-seq analysis. **b** scRNA-seq data (n = 11,327 single cells from the liver) for PBS and rIL-33-treated livers, shown as nonlinear representations of the top 50 principal components; the cells are colored based on uniform manifold approximation and projection (UMAP)-based cell clusters. **c** Dot plot showing the expression of zonation markers (*Glul* and *Cdh1*, left) and differentially expressed genes (DEG, right) in each hepatocyte cluster (n = 10,186 hepatocytes). **d**, **e** Interaction analysis using CellPhoneDB. Interaction heatmap plotting the total number of receptor and ligand interactions for the indicated cell clusters (**d**). Dot plot showing the results for the IL13-IL-13R interactions with the corresponding *P*-values and mean expression levels of IL-13 in ILC2s and IL13R in hepatocytes (**e**). **f** Representative image of immunofluorescence of PCK1-positive hepatocytes and ILC2s (CD3E-KLRG1 + ). PCK1: magenta, CD3E: red, KLRG1: green, nucleus (DAPI):

blue. In total, 24 locations were observed in 5 independent mice with IL33 treatment. **g** Enriched genes upstream of the DEGs identified between the PBS and rIL-33 groups. **h** The dot plot shows the expression of genes downstream of *Hnf4a*, including *G6pc* and *Pck1*, in all hepatocytes. **i** UMAP plots (upper panel) and violin plots (lower panel) showing *G6pc* and *Pck1* expression in all hepatocytes. **j** Volcano plots showing the DEGs between the PBS and IL-33 groups in Hepatocyte4 cells. Upregulated (red) and downregulated (blue) genes in the rIL-33-treated group. **k** Dot plot showing IL13 receptor subunit (*Il13ra1*, *Il13ra2* and *Il4ra*) and *Stat3/6* expression. The expression levels of *Il13ra1* and *Stat3* were elevated in Hepatocyte4. **l** STAT3 binding to the upstream regions of the *G6pc* and *Pck1* genes was assessed by chromatin immunoprecipitation (ChIP) with quantitative PCR (qPCR) (n = 3 per group). **m** Primary hepatocytes were transfected with small interfering RNAs (siRNAs), and 48 h later, they were subjected to RT–qPCR for *G6pc* or *Pck1* (n = 3 per group). Unpaired one-sided Student's *t*-test. $^{*}P < 0.05$; $^{**}P < 0.01$; $^{†} < 10^{-10}$; $^{††} < 10^{-20}$; $^{†††} < 10^{-30}$.

compared with the lungs and that the elevated expression of *Il13* observed in liver ILC2s could be explained, at least in part, by down-regulation of *AP-1* genes that reduced the AP-1 family-mediated suppression of GATA3 transcriptional activity (Fig. 8h).

In summary, our observations by physiological and comprehensive transcriptomic approaches combined with LC–MS/MS analysis reveal the additional function of liver ILC2s and the role of the AP-1 family in GATA3 signaling in metabolism.

## Discussion

Innate lymphoid cells are known to play roles in protecting organs from infectious diseases and exacerbating allergies. For example, ILC2s in the lungs are involved in responses to viral infection starting in early life and are mediated by IL-33 stimulation, and a stronger cytokine induction in ILC2s corresponds with exacerbation of asthma severity[5]. In the gut, ILC2s are major sources of IL-10, which protects against parasitic helminths[6]. However, a limited number of studies have examined the characteristics and roles of ILC2s in the liver, especially their involvement in glucose metabolism. In this study, we found that IL-33 stimulation suppressed gluconeogenesis in the liver through ILC2-derived *IL-13* expression. scRNA-seq revealed that *Il13* expression was more strongly induced by IL-33 stimulation in liver ILC2s than in lung ILC2s. Finally, we found that differences in the expression patterns of Gata3-binding partners resulted in differential cytokine production in the lungs, even though GATA3 is a key transcription factor for ILC2s both in the liver and in the lungs.

Insulin resistance in the liver results in excessive gluconeogenesis, and gluconeogenesis dysregulation is associated with worsened regulation of glucose metabolism, especially under obese conditions. We demonstrated that IL-13 derived from ILC2s suppressed gluconeogenesis by downregulating the expression of enzymatic genes, including *G6pc* and *Pck1*, in response to rIL-33 administration. Inoue et al. reported that Kupffer cell-derived IL-6 suppresses gluconeogenic enzymes in hepatocytes through the STAT3 signaling pathway[23]. In addition, a recent study reported that IL-13 directly suppresses gluconeogenic enzyme activity in hepatocytes via STAT3 signaling[17]. Consistent with these reports, our scRNA-seq data indicated that *Stat3* was highly expressed in hepatocytes, demonstrating a significant interaction with ILC2s and correlating with the downregulation of *G6pc* and *Pck1* and their upstream transcription factor gene *Hnf4a*. Our findings suggested that ILC2-derived IL-13 improves glucose metabolism via cellular interactions in the liver.

Numerous studies have been performed on heterogeneous populations of immune cells, especially macrophages, and the results have suggested that these populations have tissue- and niche-specific functions[24]. Transcriptional profiling of ILCs across human and mouse tissues has revealed unique features that are dependent on the tissue

microenvironment[25–30]. We elucidated the features specific to liver ILC2s via comparison with lung ILC2s. IL-13 is a prevalent cytokine found in ILC2s; however, our data showed that *Il13* transcription in response to IL-33 stimulation was significantly enhanced in the liver compared to the lungs. This finding supports the existence of inter-organ heterogeneity among ILC2s, and the transcription complex analysis indicated that the effector cytokines in ILC2s may be regulated in an organ-specific manner.

The transcription factor GATA3 is a well-known regulator of cytokine production, and we found that the expression patterns of cytokines downstream differed in ILC2s from the liver compared with ILC2s from the lungs. This diversity of downstream molecule expression was not simply associated with concomitant differences in *Gata3* expression; therefore, we examined the expression patterns of GATA3-binding partners. Our omics data, including GATA3-IP-LC–MS/MS data, suggested that GATA3 and JunB interacted in liver ILC2s. In the context of cancer immunity, tumor ILC2s express *Junb* more highly than lymph node ILC2s[31]. During helminth infections, another AP-1 family molecule, basic leucine zipper ATF-like transcription factor (BATF), induces IL-4 and IL-13 in ILC2s[32]. Our scRNA-seq data showed that AP-1 family members (*Junb*, *Fos*, and *Fosb*) were expressed at lower levels in rIL-33-treated liver ILC2s than in rIL-33-treated lung ILC2s, suggesting that AP-1 family members suppress *Il13* expression by associating with the promoter or enhancer regions of *Il13* gene loci via interaction with GATA3 in ILC2s. Although AP-1 family members are known as activators, several reports indicated that they can have suppressive functions depending on the context[33–35]. Our functional assays also suggested that some AP-1 family members can behave as suppressors in liver ILC2s in some contexts (Figs. 7i and 8b, c).

Our findings demonstrate that IL-33 stimulation in vivo drastically modulates metabolic signaling in hepatocytes through liver ILC2-derived IL-13 expression, providing insights into additional therapeutic targets for managing excessive gluconeogenesis, especially under obese diabetic conditions. We believe the mechanism uncovered in our study could introduce new avenues for the treatment of obese patients with diabetes.

## Methods

Our research complied with relevant ethical regulations and the research protocols were approved by the ethics committee for animals at Chiba University.

### Mice

BALB/c, C57BL/6, and nude mice were obtained from Japan SLC. NSG mice (#: 005557) with the NOD-SCID genetic background and carrying a null allele of the *Il2rg* gene were obtained from Charles River

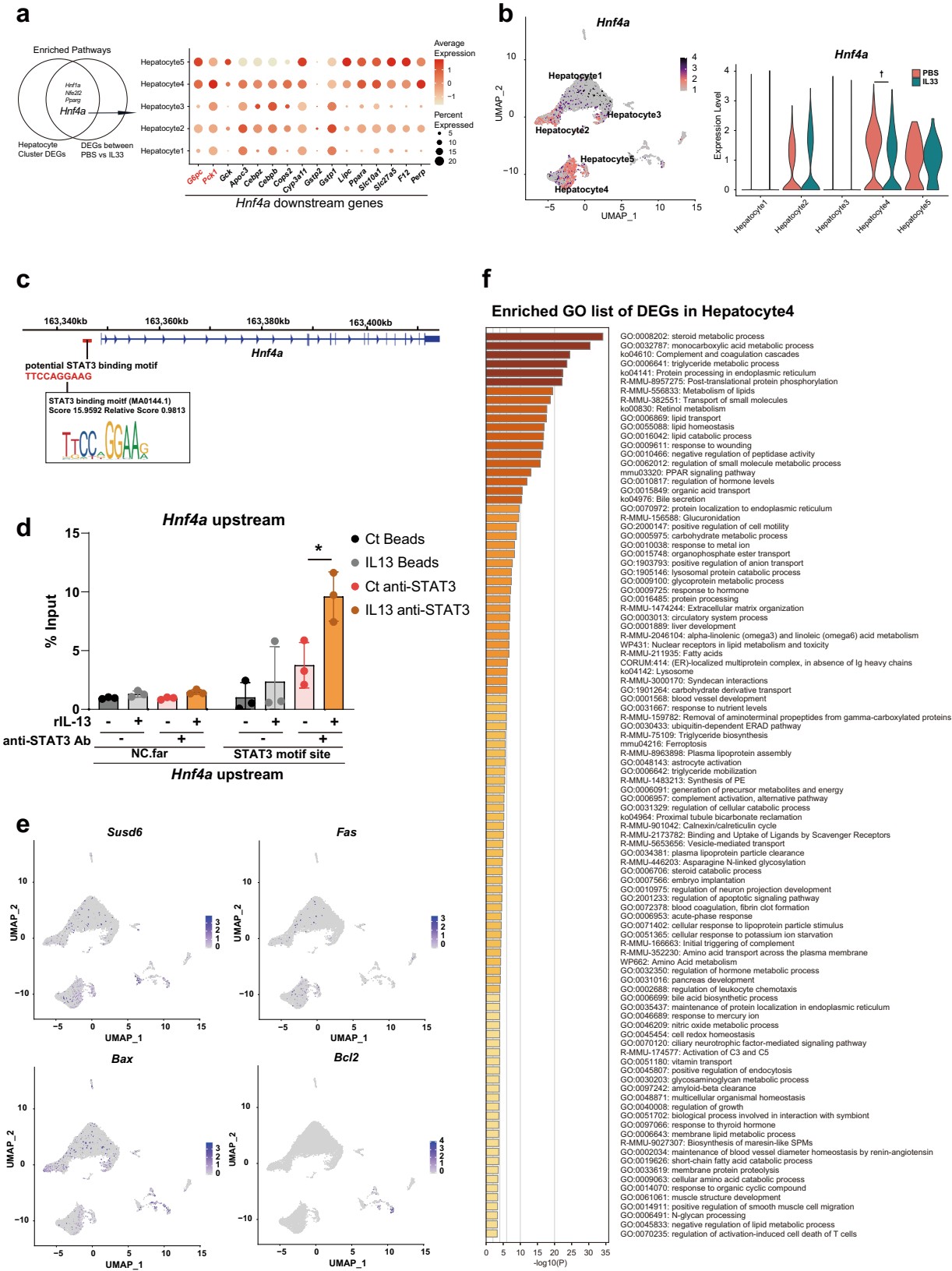

Laboratories (Japan). *Il13⁻/⁻* BALB/c mice were provided by The Medical Research Council (Cambridge, UK)[36]. A 3-month HFD model was achieved by feeding mice a dietary chow consisting of 60% kcal fat (Research Diets D12492) beginning at the age of 8 weeks and extending for a period of 5 months. All mice were allowed ad libitum access to food and water and were maintained in specific pathogen-free

conditions in a 22 °C temperature-controlled room with a 12 h light-12 h dark cycle. Humidity was maintained in the range of 40–70%. All mice were male. Unless otherwise noted, all mice were BALB/c and used for experiments at 6–8 weeks of age. The research protocols were approved by the ethics committee for animals at Chiba University (registration number: 3-161, 1-68).

**Fig. 6 | IL-13 suppresses the expression of gluconeogenic enzymes via Stat3 signaling. a** Venn diagram showing common upstream genes (*Hnf4a*, *Hnf1a*, *Nfe2l2*, and *Pparg*) of differentially expressed genes (DEG) identified by comparing clusters or treatment conditions (left panel). Dot plot showing gene expression downstream of *Hnf4a* in each cluster (right panel). **b** Uniform manifold approximation and projection (UMAP) plots (left panel) and violin plots (right panel) showing *Hnf4a* expression. **c** Potential STAT3 binding sites upstream of *Hnf4a* are shown with the genomic DNA sequence and with the motif ID, score, and relative scores analyzed by the JASPAR database. **d** STAT3 binding to the upstream regions of the *Hnf4a* gene was assessed by chromatin immunoprecipitation (ChIP) with quantitative PCR (qPCR) (*n* = 3 per group). **e** UMAP plots showing the expression of cell death markers (*Susd6*, *Fas*, *Bax* and *Bcl2*). **f** Enriched Gene Ontology (GO) term list of DEGs in the Hepatocyte4 cluster. Unpaired one-sided Student's *t*-test. $^*P < 0.05$; $^† < 10^{-10}$. Each bar and its error bars represent the mean ± SD.

## In vivo cytokine treatment

Mice were administered 200 μL of sterile PBS as a control or 0.5 μg of carrier-free murine rIL-33 (BioLegend, 580506) in 200 μL of sterile PBS via daily intraperitoneal injection for 5 days, unless otherwise noted. Each mouse was used for blood glucose measurements and euthanized the day after the final injection. An independent series of mice were administered 1 μg of carrier-free murine rIL-13 (BioLegend, 575906) in a single injection and then subjected to PTT analysis the next day. Fifty micrograms of control IgG (BioLegend, 400192) or anti-mouse IL-13 neutralizing antibody (InvivoGen, mabg-mil13-5) was administered via daily intraperitoneal injection for 4 days.

## Tolerance tests and evaluation of blood glucose and insulin levels

Before each tolerance test, the mice were fasted for 16 h with free access to drinking water. Then, 2 g/kg body weight pyruvate was administered for the PTT[37], 2 g/kg body weight glycerol was administered for the GlycerolTT[38], and 0.075–0.1 U/kg body weight insulin was administered for the ITT[37]. In each test, fasting blood glucose levels of mice were measured, after which pyruvate, glycerol, and insulin were diluted in PBS and injected at the appropriate volume i.p. One series of loading tests was performed on day 1. Mice were subjected to PTT, and then the GlycerolTT or ITT was performed 1 or 3 days after PTT, with daily rIL-33 injections. Changes in blood glucose levels were measured during 0−60 min (PTT, GlycerolTT) or 0−240 min (ITT). All reagents utilized for these tests were purchased from Sigma−Aldrich.

## Insulin measurements

Blood samples from mice (wt BALB/c, nude, and NSG mice) fasted for 16 h were collected from the retro-orbital vein. Insulin concentrations were measured using an ultrasensitive mouse insulin enzyme-linked immunosorbent assay (ELISA) (Morinaga, M1108).

## IL-13 ELISA

Serum IL-13 levels were quantified using an ELISA kit (R&D Systems, DY413-05) according to the manufacturer's instructions.

## Glycogen content assay

The glycogen content in liver tissues was quantified using a glycogen content assay kit (Abcam) according to the manufacturer's instructions. Liver tissue (<100 mg) was washed with cold PBS and ddH₂O, homogenized, and boiled for 10 min to inactivate enzymes. After centrifugation for 10 min at 4 °C at 18,000 × *g*, the supernatant was collected and treated with reagents, the optical density was measured at 570 nm, and the glycogen content was calculated.

## Immunofluorescence staining

Liver samples were harvested and fixed in 4% paraformaldehyde (PFA) overnight at 4 °C. The cryosections of tissues were permeabilized in PBS with 0.1% Triton X-100 and blocked for 30 min at room temperature (RT) in 5% normal goat serum. After eliminating the background signal with Endogenous Biotin-Blocking Kit (Invitrogen, E21390), immunofluorescence staining was performed with the following antibodies: anti-KLRG1 (1:25, FITC conjugated, Biolegend, 138409), rabbit anti-PCK1 (1:50, Abcam, ab70358) and anti-CD3e (1:25, biotin conjugated, eBioscience, 13-0033-82). The secondary antibodies were an

Alexa Fluor 647 goat anti-rabbit antibody (1:50, Molecular Probes, A-21244) and Alexa Fluor 594 streptavidin (1:50, Jackson ImmunoResearch, 016-580-084). Primary antibody staining was performed at 4 °C overnight, and secondary antibody staining was performed at RT for 2 h. DAPI was used for nuclear staining. The immunofluorescence images were captured with Zeiss LSM 980 confocal microscope (microscope: Axio Observer7, objective lens: ×20 Plan Apochromat) and analyzed with Zeiss ZEN Blue software. Pseudocolored magenta was used for Alexa Fluor 647 dye, and orthogonal projection was performed to represent a 12 μm-section in two dimensions.

## Primary hepatocyte isolation and culture

Primary hepatocytes were isolated from 8- to 10-week-old mice via a two-step collagenase digestion method[39]. Specifically, mice were euthanized and sequentially perfused with ethylene glycol-bis(β-aminoethyl ether)-N,N,N,N-tetraacetic acid (EGTA) and collagenase (Gibco) starting from the inferior vena cava and draining through the portal vein. Single-cell suspensions were washed with Dulbecco's modified Eagle's medium (DMEM), and 1 × 10⁵/cm² cells were plated in William's E medium (Thermo Fisher Scientific, 12551032) supplemented with 10% fetal bovine serum and 1% penicillin−streptomycin.

## Treatment of primary hepatocytes

Primary hepatocytes were incubated with or without 10 ng/mL rIL-13 for 2 h at 37 °C and then stimulated with 100 μM cAMP or 100 nM glucagon and with or without 10 ng/mL rIL-13 for 3 h at 37 °C. Cells were assigned to one of the following five groups: the control (vehicle), cAMP-treated (cAMP), cAMP- and rIL-13-treated (cAMP+rIL-13), glucagon-treated (Gluca), and glucagon- and rIL-13-treated (Gluca+rIL-13) groups.

## Sampling of organ ILC2s and cell sorting

Male *Il13*⁺/⁺, *Il13*⁺/⁻ and *Il13*⁻/⁻ BALB/c mice received daily injections of rIL-33 (0.5 μg, i.p.) for 5 days. The mice were euthanized and perfused with PBS, and then liver and lung ILC2s were harvested. Briefly, the mouse livers and lungs were digested using a gentle MACS Octo Dissociator with Heaters (Miltenyi Biotec) using a liver dissociation kit (Miltenyi, 130-105-807, program: 37C_m_LIDK_1) and a lung dissociation kit (Miltenyi, 130-095-927, m_lung_01, incubation at 37 °C for 30 min, and then m_lung_02 program), respectively. Debris was removed using the protocol published by Itoh, with some modifications[40]. After filtration through a 70-μm cell strainer, the cells were collected by centrifugation at 500 × *g* for 5 min at 4 °C and resuspended in 40% Percoll Plus (GE Healthcare). Then, 75% Percoll was gently added below the 40% Percoll layer. After centrifugation at 700 × *g* for 20 min at RT, the cells at the interface were collected in PBS containing 1% FBS and 2 mM EDTA, and lineage-positive cells were depleted using MACS (Miltenyi) with an anti-mouse lineage panel (BioLegend, 133307) and streptavidin nanobeads (BioLegend, 480016). Then, single-cell suspensions were stained using blocking and staining antibodies in the dark at 4 °C. Liver Lin⁻Th1.2⁺IL-7Rα⁺ST2⁺ ILC2s were sorted using a FACSAria cell sorter (BD Biosciences) and cultured or directly used for experiments.

## ILC2 in vitro cultures

ILC2 culture was performed as before[41]. ILC2s isolated from each organ were cultured with complete medium (RPMI-1640 medium

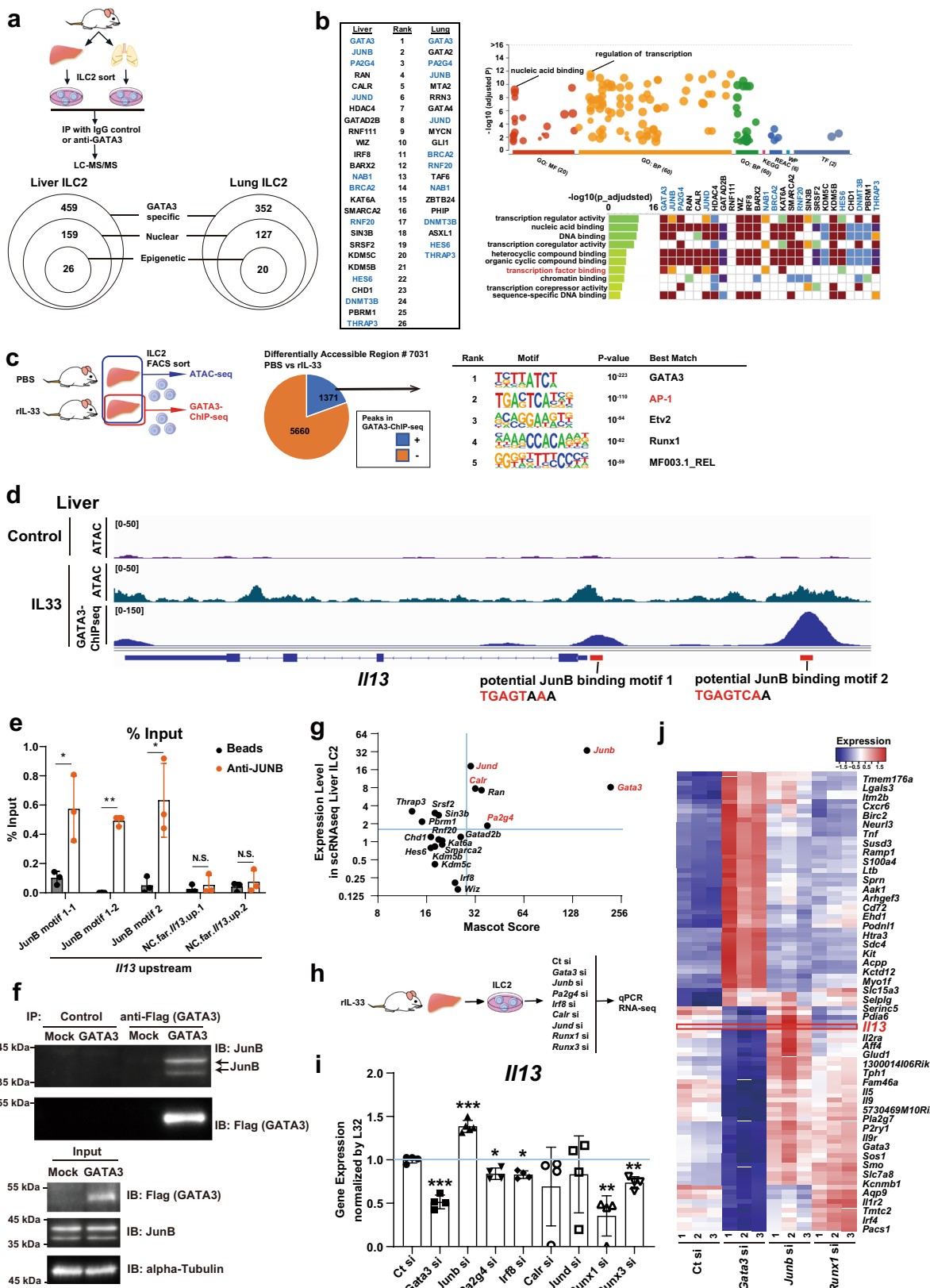

[Sigma–Aldrich], 10 mM HEPES [Life Technologies], 1 mM sodium pyruvate [Life Technologies], 1% MEM nonessential amino acids [Life Technologies], 0.1% 2-mercaptoethanol and L-glutamine) with 25 U/mL rIL-7Rα (BioLegend), 10 ng/mL rIL-25 (BioLegend), and 10 ng/mL rIL-33. The ILC2s were maintained at 37 °C with 5% CO₂ and passaged after 2–3 days into the same volume of medium containing cytokines. The ILC2 cell line ILC2/b6 was generously donated by Dr. Q Yang. The ILC2/b6 cells were cultured in OP9 medium (α-MEM, 20% FBS, 50 μM β-mercaptoethanol, and Pen-Step-Glutamine) supplemented with 10 ng/mL IL-2, IL-7, and IL-33 (PeproTech Inc.)[42].

**Fig. 7 | AP-1 family members bind to GATA3 in liver and lung group 2 innate lymphoid cells (ILC2) and suppress GATA3 function to induce *Il13* expression.** **a** Experimental schema. Sorted liver and lung ILC2s were cultured and then subjected to immunoprecipitation with an IgG control or anti-GATA3 antibody and then to liquid chromatography–mass spectrometry/mass spectrometry (LC –MS/MS) analysis. The Mascot score was evaluated, and GATA3-specific interacting proteins were identified. **b** List of GATA3-specific nuclear and epigenetic factors. Bubble plot and heatmap showing the functions and Gene Ontology (GO) terms of the listed proteins. **c** Experimental schematic of assay for transposase-accessible chromatin with high-throughput sequencing (ATAC-seq) and GATA3 chromatin immunoprecipitation sequencing (ChIP-seq). Differentially accessible regions (DAR) between recombinant (rIL-33)- and phosphate-buffered saline (PBS)-treated ILC2s were classified into GATA3-bound (blue) and nonbound (orange) regions (middle). Enriched sequence motifs for the GATA3-bound DARs (right). **d** Representative tracks at an arbitrary locus (near *Il13*) illustrating signals in ATAC-seq (first and second tracks) and GATA3 ChIP-seq (third track) samples of liver ILC2s. **e** JunB binding to the upstream regions of the *Il13* gene was assessed by ChIP with quantitative PCR (qPCR) (right) (*n* = 3 per group). The results show JunB

binding motif1 (first and second column), motif2 (third), and nonbinding control regions far upstream of the *Il13* gene (fourth and fifth). **f** The ILC2 cell line ILC2/b6 was infected with a retrovirus encoding FLAG-GATA3 and a retrovirus encoding JunB. Five days later, the extracts were subjected to immunoprecipitation (IP) with an anti-FLAG (GATA3) antibody and then subjected to immunoblotting (IB) with an anti-JunB antibody (upper panel). The total cell lysates (input) were also subjected to IB in parallel (lower panel). These are representative figures of two independent experiments. **g** Mascot score and expression levels from single-cell RNA sequencing (scRNA-seq) data in liver ILC2s. **h** Experimental schema. Twenty-four hours after small interfering RNA (siRNA) transfection, liver ILC2s were subjected to real-time quantitative PCR (RT–qPCR) and RNA-seq analyses. **i** RT–qPCR of liver ILC2s transfected with various siRNAs. The expression of *Il13* is shown (normalized to *L32*) (Ct si: *n* = 4, *Gata3* si: *n* = 4, *Junb* si: *n* = 5, *Pa2g4* si: *n* = 4, *Irf8* si: *n* = 4, *Calr* si: *n* = 4, *Jund* si: *n* = 4, *Runx1* si: *n* = 4, *Runx3* si: n = 4). **j** Heatmap showing differentially expressed genes (DEG) in liver ILC2s transfected with Ct si, *Gata3* si, *Junb* si, or *Runx1* si. Unpaired one-sided Student's *t*-test. *$^*P < 0.05$; *$^{**}P < 0.01$; *$^{***}P < 0.001$. Each bar and its error bars represent the mean ± SD.

## Coculture of primary hepatocytes and hepatic ILC2s
Primary hepatic ILC2s were cultured for several days. Then, the ILC2s were washed with PBS, and $2 \times 10^4/cm^2$ ILC2s were plated on top of a monolayer of primary hepatocytes in the presence or absence of 5 μg/mL anti-mouse IL-13 neutralizing antibody (InvivoGen, mabg-mil13-5). After 6 h of culture, floating ILC2s were aspirated and washed with PBS, and hepatocytes were immediately subjected to RNA extraction or metabolomic analysis.

## Metabolomic analysis
Primary hepatocytes were washed with 5% (w/w) mannitol and immersed in methanol, and extracts were subjected to ultrafiltration and frozen immediately. CE-TOFMS and CE-QqQMS (C-Scope) were performed to evaluate central energy metabolism in hepatocytes (Human Metabolome Technologies, Japan).

## ILC2 transfer into NSG mice
Liver ILC2s were purified from wt BALB/c mice that were treated with rIL-33 for 5 consecutive days and then cultured for 3–5 days. After culturing, $5 \times 10^5$ to $1 \times 10^6$ liver ILC2s were suspended in 300 μL of PBS and then intravenously injected into NSG mice. Control NSG mice were intravenously injected with 300 μL of PBS. The control and recipient mice received rIL-33 injections every day, and fasting blood glucose was measured after 3 and 5 days.

## Flow cytometric analysis
Cells were incubated with an anti-CD16/CD32 monoclonal antibody (mAb, eBioscience, 14-0161-82, 93, 1:33) and then stained with the following dye or anti-mouse antibodies: viability dye APC-Cy7 (eBioscience, 65-0865-14), anti-lineage cocktail-FITC (BioLegend, 133302, 145-2C11; RB6-8C5; RA3-6B2; Ter-119; M1/70, 1:40), anti-TCRβ-FITC (BioLegend, 109206, H57-597, 1:200), anti-CD90.2-PE/Cy7 (BioLegend, 105326, 30-H12, 1:200), anti-CD127-APC (BioLegend, 135012, A7R34, 1:100), anti-ST2-BV421 (BioLegend, 145309, DIH9, 1:100), anti-CD3E-FITC (BioLegend, 100203, 17A2, 1:200), anti-TCRg-PE (BioLegend, 118107, GL3, 1:100). Gating was performed as follows: ILC2s were defined as live Lin⁻ (CD3e⁻Ly6G⁻Ly6c⁻CD11b⁻CD45R⁻TER119⁻TCRb⁻) CD127⁺Thy1⁺ST2⁺ cells. Stained cells were examined with a FACSCanto II flow cytometer (BD Biosciences), and the results were analyzed using FlowJo software version 10.4.2 (TreeStar, FlowJo).

## Intracellular cytokine staining
Freshly isolated lymphocytes from livers and lungs were restimulated with phorbol 12-myristate 13-acetate (50 ng/mL) and ionomycin (500 ng/mL) for 4 h in the presence of monensin (2 μM) and then

stained for ILC2 surface markers. The cells were fixed with 4% PFA for 10 min at RT and permeabilized in permeabilization buffer (50 mM NaCl, 5 mM EDTA, 0.02% NaN₃, pH 7.5) containing 0.5% Triton X-100 for 10 min on ice. After blocking with 3% bovine serum albumin in PBS for 15 min, the cells were stained on ice for 30 min with anti-IL-13 (BioLegend, 159403, W17010B, 1:100). Flow cytometry was performed with a FACSCanto II (BD Biosciences) or FACSymphony A3 (BD Biosciences) flow cytometer, and the results were analyzed with FlowJo.

## Real-time PCR
Real-time PCR was performed as previously described[43]. Liver tissues were collected from mice soon after euthanasia and then homogenized using a TissueRuptor II system (Qiagen). RNA was extracted using an RNeasy mini kit (Qiagen, 74104), and its quantity and quality were evaluated with a NanoDrop spectrophotometer (Thermo Fisher Scientific). The RNA was then reverse-transcribed into cDNA using a ReverTra Ace qPCR RT kit (Toyobo), and the resulting cDNA was amplified by RT–PCR using Fast SYBR Green (Thermo Fisher Scientific, 4385614). Ct values were quantified using a StepOnePlus Real-Time PCR system (Applied Biosystems) with the default fast mode setting (stage 1; 95.0 °C for 20 s, stage 2; 40 cycles at 95.0 °C for 1 s, and 60.0 °C for 20 s). Relative gene expression was quantified using the $2^{-\Delta\Delta CT}$ method and normalized to *Rpl32* expression. The genes and primer pairs are listed in Supplementary Table 5.

## siRNA or plasmid transfection
Cultured ILC2s were electroporated with a NEON transfection kit and device (Thermo Fisher Scientific) using three pulses of 1800 V for 10 ms to transfect select siRNAs or pCMV-myc-JunB vector. After 24 h of transfection, cells were collected and used for RT–qPCR or RNA-seq. Primary hepatocytes were transfected with RNAi using Lipofectamine RNAi Max (Thermo Fisher, 13778075) according to the manufacturer's instructions. The siRNAs were as follows: Ct si (Negative Control #1, Cat# AM4635) and Target si (Cat# 4390771, 4390815). The IDs were as follows: Gata3 si: s61810, Junb si: s68567, Pa2g4 si: s201806, Calr si: s63270, Jund si: s201553, Runx1 si: s201126, Runx3 si: s63447, Il4ra si: s68281, Il13ra1 si: s68211, Stat3 si: s74452, Stat6 si: s74463. The pCMV-myc-JunB vectors used were described previously[44].

## GATA3-IP and subsequent LC–MS/MS analysis
The protocol was performed as before[20,21,45,46]. Liver ILC2s ($1 \times 10^7$) and lung ILC2s ($1.8 \times 10^6$) were solubilized with the following protease inhibitor-containing IP buffer: 50 mM Tris-HCl (pH 7.5), 150 mM NaCl, 10% glycerol, 0.1% Tween, 1 mM EDTA, 10 mM NaF, 1 mM DTT, and a protease inhibitor cocktail (Roche Applied Science). The cells were

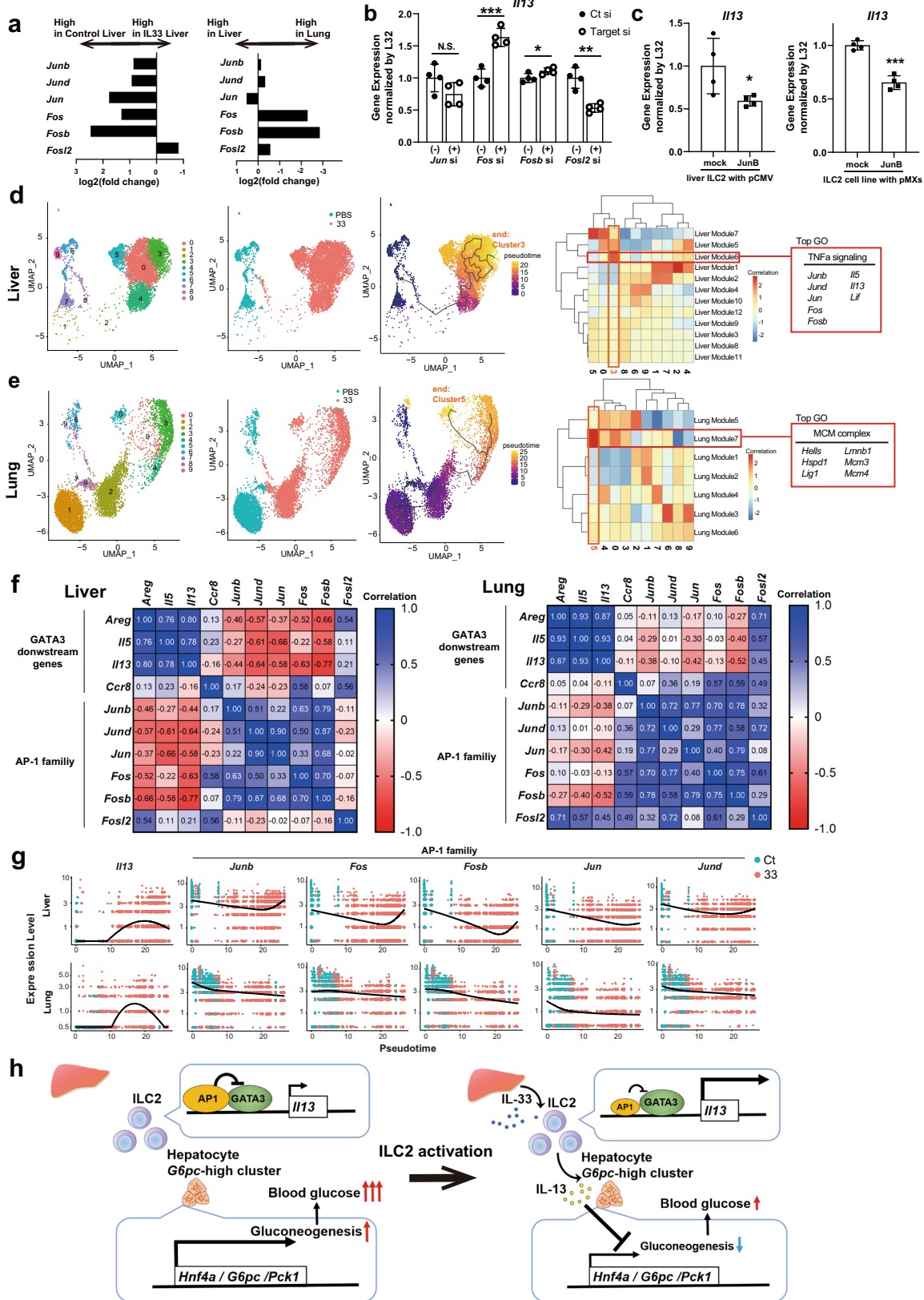

lysed on ice for 30 min with gentle shaking and sonicated on a Misonix S-4000 sonicator (Qsonica) for three cycles at an amplitude of 20 for 30 s followed by 30 s of rest. Protein complexes were subjected to IP with or without an anti-GATA3 antibody (Santa Cruz Biotechnology, sc-268; R&D Systems, MAB26051) and then separated by SDS–PAGE. The bands were excised from the gel, and the gel pieces were washed twice with 100 mM bicarbonate in acetonitrile. The proteins were then digested with trypsin for 8 h at 37 °C. After adding 0.1% formic acid to the supernatant, the peptides were analyzed by LC–MS/MS with an Advance UHPLC (Bruker) and an Orbitrap Velos Pro Mass Spectro-meter (Thermo Fisher Scientific). The resulting LC–MS/MS dataset was analyzed using the Mascot software program (version 2.5.1, Matrix

**Fig. 8 | Trajectory analysis reveals that AP-1 family member expression negatively correlates with *Il13* expression, especially in liver group 2 innate lymphoid cells (ILC2s). a–g** Analysis based on single-cell RNA sequencing (scRNA-seq) data for liver ILC2s and lung ILC2s from phosphate-buffered saline-treated or recombinant IL-33 (rIL-33)-treated wild-type BALB/c mice. **a** Fold-change in the gene expression of AP-1 family members between control liver ILC2s and rIL-33-treated liver ILC2s (left) and between liver ILC2s and lung ILC2s (right). **b** Real-time quantitative PCR (RT–qPCR) analysis of *Il13* 24 h after siRNA transfection of AP-1 family members (*Jun*, *Fos*, *Fosb*, *Fosl2*) in liver ILC2s (normalized to L32) (n = 3 per group). **c** RT–qPCR analysis of *Il13* 24 h after transfection with a pCMV vector encoding JunB into liver ILC2s (left) and 5 days after infection with a retrovirus encoding JunB in the ILC2 cell line ILC2/b6 (right) (n = 3 per group). **d, e** Trajectory analysis of liver ILC2s and lung ILC2s using Monocle3. The root is defined as control ILC2s. **d** Module analysis showed that AP-1 family members and some genes downstream of GATA3 (*Il13* and *Il5*) were coregulated and strongly associated with the end cluster (cluster 3). **e** Trajectory analysis of lung ILC2s showed a different end cluster compared with liver ILC2s, which was not associated with AP-1 family members and GATA3 downstream genes (cluster 5). **f** Correlation matrix for AP-1 family members and *Gata3* downstream genes based on the average gene expression of each cluster in liver ILC2s (left) and lung ILC2s (right). **g** Pseudotime plot showing the expression levels of *Il13* and AP-1 family members along the pseudotime axis. **h** Graphical abstract for this study. IL-33 induces IL-13 expression in liver ILC2s. IL-13 directly suppresses gluconeogenesis in hepatocytes by inhibiting gluconeogenic enzymes, such as *G6pc* and *Pck1*, leading to reduced blood glucose levels. AP-1 family members bind to GATA3 and suppress IL-13 production in liver ILC2s. Unpaired one-sided Student's *t*-test. *P < 0.05; **P < 0.01; ***P < 0.001. Each bar and its error bars represent the mean ± SD.

Science). The Mascot score indicated the probability that the observed match was a random event (a Mascot score > 100 means the absolute probability is <1e⁻¹⁰). Proteins with significant Mascot scores with a high signal/noise ratio (anti-GATA3/IgG control > 1) were considered specific GATA3-interacting proteins.

### Retrovirus infection and subsequent FLAG-IP-immunoblotting

ILC2/b6 cells were infected with either a mock control (pMxs-IRES-hNGFR) or a Myc-Flag-GATA3-containing retrovirus[47,48]. Five days after infection, Myc-Flag-tagged GATA3-infected hNGFR⁺ cells were solubilized with the following protease inhibitor-containing IP buffer: 50 mM Tris-HCl (pH 7.5), 150 mM NaCl, 10% glycerol, 0.1% Tween, 1 mM EDTA, 10 mM NaF, 1 mM DTT and a protease inhibitor cocktail (Roche Applied Science). The cells were lysed on ice for 30 min with gentle shaking and sonicated on a Misonix S-4000 sonicator (Qsonica) for 3 cycles at an amplitude of 20 for 30 s followed by 30 s of rest. The insoluble materials were removed by centrifugation, and IP with 50 μL anti-Flag M2 agarose (Sigma–Aldrich, A2220) was performed overnight at 4 °C. The immune complexes were subjected to immunoblotting with anti-JunB (Sant Cruz, sc8051, 1 μg/mL), anti-Flag (Sigma, M2, A2220, 1 μg/mL) and anti-Tubulinα (Sigma, T6119, 1 μg/mL). Image data were obtained using Fusion Solo S (Vilber).

### GATA3-ChIP-seq and data analysis

ChIP-seq analyses were performed as reported previously[49]. Sorted liver ILC2s (1.7 × 10⁶) and liver ILCs (0.5 × 10⁶) were fixed with 1% formaldehyde (Thermo Fisher Scientific, 28906) in MEM alpha (Thermo Fisher Scientific, 12561-056) for 10 min at RT. The reaction was quenched by the addition of a 1/10 volume of 0.125 M glycine, incubated for 5 min at RT, washed with PBS, and centrifuged at 500 × g. The pelleted nuclei were resuspended in lysis buffer (0.5% SDS, 10 mM EDTA, 0.5 mM EGTA, 50 mM Tris-HCl [pH 8], and protein inhibitor cocktail) and sonicated on a Bioruptor (Diagenode) for 18 cycles of 30 s of sonication followed by 30 s of rest at maximum power. Per 10⁷ cells, 5 μg of antibody was hybridized to Dynabeads M-280 sheep anti-rabbit, Dynabeads M-280 sheep anti-mouse, or Dynabeads Protein A/G (Invitrogen) and incubated with diluted chromatin complexes overnight at 4 °C. The beads were then washed five times with IP buffer and eluted for 6 h at 65 °C in ChIP elution buffer (20 mM Tris-HCl, pH 7.5, 5 mM EDTA 50 mM NaCl, 1% SDS, and 50 mg proteinase K). The precipitated chromatin fragments were cleaned using ChIP DNA Clean & Concentrator (Zymo Research) and subjected to qPCR on a 7900HT Fast Real-Time PCR System (Applied Biosystems) with SYBR GreenER qPCR SuperMix (Invitrogen). Mouse anti-Gata3 mAbs (a mixture of sc-268 from Santa Cruz Biotechnology and MAB26051 from R&D Systems) were used. ChIP-seq libraries were constructed using an NEBNext ChIP-Seq Library Preparation Kit (Illumina, San Diego, CA, USA; NEB #E6240) according to the manufacturer's instructions. Deep sequencing was performed on the Illumina NextSeq 500 platform using a TruSeq Rapid SBS Kit (Illumina) in 60-base single-end mode according to the manufacturer's protocol. The annotation of enhancers for the nearest genes was performed with GREAT (http://bejerano.stanford.edu/great/public/html/index.php). A heatmap of the ChIP-seq peaks was produced using HOMER for enrichment calculations, and IGV was used for visualization. Reads were mapped to the mouse genome assembly version from December 2011 (mm10) by using the Bowtie2 mapping tool with a default setting. Peaks were called using MACS2[50] software, and those identified in both replicates were used for further analyses (Fig. 7c). Data quantification was conducted using the "annotatePeaks.pl" routine in HOMER[51]. Motif analysis was performed using the "findMotifsGenome.pl" routine in HOMER.

### ChIP-PCR

Cultured liver ILC2s (5 × 10⁶) were incubated with phorbol 12-myristate 13-acetate (50 ng/mL) and ionomycin (500 ng/mL) for 3 h and then crosslinked. ChIP was performed as in the protocol for ChIP-seq with some modifications. Cells were sonicated on Q700 (QSONICA) for 20 to 25 cycles of 15 s of sonication followed by 45 s of rest at an amplitude of 60 (total: 6000J). Mouse anti-JunB mAbs (a mixture of 10 μL [1:50] of sc-8051 from Santa Cruz Biotechnology and 10 μL [1:50] of C37F9 [#3753] from Cell Signaling Technology) were used. Primary hepatocytes were isolated from wt BALB/c mice and cultured with serum-free DMEM in the presence of 100 μM cAMP with or without 10 ng/mL rIL-13 for 3 h at 37 °C. Next, 1 × 10⁷ harvested hepatocytes were processed using a SimpleChIP kit (Cell Signaling Technology, #9002). Anti-STAT3 mAbs (a mixture of 5 μL [1:100] of 124H6 [#9139] and 10 μL [1:50] of D3Z2G [#12640] from Cell Signaling Technology) were used. RT–PCR was performed using eluted DNA.

### ATAC-seq

ATAC-seq analyses of sorted liver and lung ILC2s were performed as before, with minor modifications[52]. A total of 5,000–50,000 cells were pelleted and washed with 50 μL of 1× PBS before treatment with 50 μL of 2× lysis buffer. The nuclei were resuspended in 40 μL of Tagment DNA enzyme buffer supplemented with 2.5 μL of Tn5 transposase (Illumina, 15027865) to tag and fragment the accessible chromatin. The mixture was incubated at 37 °C for 30 min. The DNA fragments were purified with a MinElute kit (Qiagen, 28004) and amplified by 11–13 PCR cycles based on the amplification curve. The libraries were sequenced for 60 cycles (single-end reads) on an Illumina HiSeq 1500 instrument. The reads were mapped to the mouse genome assembly version from December 2011 (GRCm38/mm10) by using the Bowtie2 mapping tool with a default setting. Peaks were called using MACS2 software with the broad option, and those identified in both replicates were used for further analyses (Fig. 7c). Data quantification was conducted using the "annotatePeaks.pl" routine in HOMER. DARs were detected using the "getDifferentialPeaksReplicates.pl" routine in HOMER.

## RNA-seq and data analysis

Total RNA was purified from 30,000–100,000 liver ILC2s and 100,000 primary hepatocytes using an RNeasy Plus Micro Kit (Qiagen). The RNA concentration and RNA integrity were measured using an Agilent 2100 bioanalyzer (Agilent). Library construction and sequencing were performed as previously described[53]. Briefly, libraries for RNA-seq were prepared using a TruSeq RNA Sample Prep Kit (Illumina, 20020594) in accordance with the manufacturer's protocol. Sequencing was performed on the Illumina HiSeq 1500 or NextSeq 500 platform using a TruSeq Rapid SBS Kit (Illumina, FC-402-4002) in the 50-base single-end mode, according to the manufacturer's protocol. The TopHat program (version 1.3.2 with default parameters) was used to align reads to a mouse reference genome (mm10). Then, gene expression values were calculated as reads per kilobase of exon unit per million mapped reads using the Cufflinks program (version 2.0.2). DEGs with a false-discovery rate of less than 0.05 were identified using an edgeR software package (version 3.28.1). A heatmap and dendrogram were also drawn using R software. Upstream gene (transcriptional regulatory relationships unraveled by Sentence-based Text-mining, TRRUST), GO, and Kyoto Encyclopedia of Genes and Genomes (KEGG) terms were used to classify the DEG functions using Metascape.

## scRNA-seq library preparation and sequencing

Single-cell suspensions of immune cells from the livers or lungs of wt 8-week-old BALB/c mice were prepared by digestion, Percoll enrichment, and the sorting method as stated above (gating: Lin⁻ [CD3e⁻Ly6G⁻Ly6c⁻CD11b⁻CD45R⁻TER119⁻TCRb⁻] CD127⁺Thy1⁺ST2⁺). Single-cell hepatocytes were also prepared using the collagenase perfusion method described above. Liberase (Gibco) was substituted for collagenase to perform milder digestion. The scRNA-seq libraries were prepared according to 10× Genomics specifications (10× Chromium Next GEM single-cell 3′ reagent kits and the library construction kit). Briefly, cell suspensions were loaded onto the 10× Genomics Chromium Controller to generate gel beads in emulsion (GEM). After GEM generation, the samples were incubated at 53 °C for 45 min in a thermal cycler (Veriti, Applied Biosystems) to generate polyA cDNA barcoded at the 5′ end by the addition of a template switch oligo linked to a cell barcode and unique molecular identifiers (UMI). The GEMs were broken, and the single-stranded cDNA was cleaned up with DynaBeads MyOne Silane Beads (Thermo Fisher Scientific). The cDNA was amplified (98 °C for 3 min, 11 cycles of 98 °C for 15 s and 63 °C for 20 s, and 72 °C for 1 min), and cDNA quality was assessed using an Agilent TapeStation. The cDNA was enzymatically fragmented, end-repaired, A-tailed, subjected to double-sided size selection with SPRIselect beads (Beckman Coulter), and ligated to adaptors provided in the kit. A unique sample index for each library was introduced through 14 cycles of PCR amplification using the indices provided in the kit (98 °C for 45 s; 14 cycles of 98 °C for 20 s, 54 °C for 30 s, and 72 °C for 20 s; 72 °C for 1 min; and a hold at 4 °C). The indexed libraries were subjected to a second round of double-sided size selection, and the libraries were then quantified and quality-assessed with an Agilent TapeStation. The libraries were submitted to GENEWIZ and clustered using a NovaSeq 6000 on a paired-end read flow cell. Sequencing was performed on R1 (10× barcode and UMI), I7 index (8 cycles; sample index) and R2 (89 cycles; transcript). The 10× Genomics Cell Ranger Single Cell Software was used for sample demultiplexing, alignment to the mouse reference genome (mm10), filtering, UMI counting, single-cell 3′-end gene counting, and quality control according to the manufacturer's parameters.

## scRNA-seq data analysis

The R package Seurat (v.3.1.2 and v4.0.4)[54] was used to cluster cells in a merged matrix. Several quality control steps were performed based on the plot pattern of the expression gene number and ratio of mitochondrial RNA in each sample. First, immune cells including doublets and dead cells were removed with flow cytometry, and those expressing fewer than 500 genes (low quality) were excluded from further analysis. Next, primary hepatocytes in which libraries were directly constructed without flow cytometry that expressed more than 3000 genes (potential doublets) and fewer than 200 genes or >50% mitochondrial genes (low quality) were first filtered out. The threshold for hepatocytes was determined by examining the validity of each mitochondrial cutoff (10–70%) because mitochondrial gene expression is constitutively active in hepatocytes due to the roles of mitochondria in energy metabolism. A strict cutoff (threshold of >10% mitochondria) eliminated enzymatically active cell populations, as reported by MacParland et al., who also used a threshold of >50% mitochondrial genes for human hepatocyte scRNA-seq[55]. The gene counts for each cell were divided by the total gene counts for each cell and multiplied by a scale factor of 10,000, after which a natural log transformation was applied. The FindVariableFeatures function was used to select variable genes with default parameters. The ScaleData function was used to scale and center the counts in the dataset. We excluded several genes for clustering and listed them in Supplementary Table 7. To define the cell types of all cells, we performed principal component analysis on the variable genes, and 50 principal components were used for cell clustering (resolution = 5) and UMAP dimensional reduction[56]. For further ILC2 subclustering, a resolution of 0.5 was used. The cluster markers were found using the FindAllMarkers function, and cell types were manually annotated based on the cluster markers. DEGs between two groups were defined as those with values of $P < 0.05$ as determined by the Welch $t$-test. Cluster markers defined by FindAllMarkers were used as DEGs of each cluster. These DEGs were used for functional analyses using the GO and KEGG pathway databases. The single-cell gene expression data were visualized with UMAP overlays, dot heatmaps, and violin plots. A volcano plot was constructed in R. We illustrated the cell cycle phase (G1, S, G2M) using the CellCycleScoring function based on cell cycle-related genes. To estimate the stage of maturation, a pseudotime trajectory analysis was performed using Monocle3[57]. To calculate the correlations of clusters and gene sets, we grouped coregulated genes into modules using Louvain community analysis with Monocle3.

## Cell–cell ligand–receptor interaction analysis

CellPhoneDB (v.2.0.0) was used to perform ligand–receptor analysis of cell populations in the liver, especially for assessment of the interactions between hepatocytes and ILC2s[58,59]. The raw counts and cell-type annotation for each cell were inputted into CellPhoneDB to determine potential ligand–receptor pairs. Pairs with $P > 0.05$ were not included in further analyses. The numbers of interactions between each pair of clusters (hepatocyte clusters 1–5 and ILC2 clusters 0–9) were plotted using downsized cells (500 cells in total).

## Analysis with g:GOSt

DEGs identified in siRNA-transfected ILC2s were included in a functional enrichment analysis query and then analyzed using g:GOSt functional profiling[60].

## Statistics and reproducibility

The data were analyzed with GraphPad Prism 9 (GraphPad Software) and are expressed as the mean ± standard deviation (SD). Significant differences between two groups were determined by unpaired one-sided $t$-test. Comparisons of more than three groups were performed using one-way analysis of variance followed by Sidak's or Tukey's multiple comparisons test. Significance was set at $P < 0.05$. The experimental findings were reliably reproduced with at least two independent experiments in Fig. 1a–g, i, j; Fig. 3h, i; Fig. 4a–e, g, i; Fig. 5f, l, m; Fig. 6d;

Fig. 7e, f, i; Fig. 8b, c; Supplementary Fig. 1a–d; Supplementary Fig. 2a–h; Supplementary Fig. 3b; Supplementary Fig. 4d. The micrograph of immunofluorescence in Fig. 5f is representative of 24 locations of 5 independent mice. Western blotting in Fig. 7f is representative figures of two independent experiments. We performed scRNA-seq of ILC2s using pooled tissues of 2–6 mice in each group (Fig. 3). ATAC-seq and ChIP-seq (Fig. 7c, d and Supplementary Fig. 6c–e) were performed with duplicates only except for GATA3-ChIP-seq from IL-33 treated lung (Supplementary Fig. 6c right, 6e). RNA-seq (Fig. 7j) of cultured ILC2s were performed with triplicates.

### Reporting summary
Further information on research design is available in the Nature Research Reporting Summary linked to this article.

### Data availability
The raw and processed data generated in this study have been deposited in the Gene Expression Omnibus (GEO) database under accession code GSE189287. The source data are provided as a Source Data file. Source data are provided with this paper.

### Code availability
The code will be made available upon reasonable request.

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

## Acknowledgements

We are grateful to Y. Endo for helpful suggestions and experimental support throughout this work, M. Mastumoto and K. Nakayama for LC–MS/MS analysis, and A. Kaneda and A. Okabe for helpful suggestions on ChIP-seq analysis and data interpretation. The authors would also like to thank R. Yuki for helpful support for ATAC-seq analysis, K. Murata for supporting scRNA-seq library construction, M. Nakanishi for basic suggestion on scRNA-seq analysis, K. Sugaya for cell sorting, Y. Sano and H. Miya for sample preparation. This work was supported by grants from the Ministry of Education, Culture, Sports, Science and Technology (Japan) (Grants-in-Aid: for Scientific Research [S] 26221305, JP19H05650, [B] #21H02974, #19H03708, #22300325; [C] #22K08644, #22K07205, #22K08619, #21K08524, #20K08397, #20K07561, #19K07635, 19K08972, #18K07439, #18K08464; Challenging Research [Exploratory] #21K19398; Early-Career Scientists #20K17527; Japan Agency for Medical Research and Development, AMED [No JP21ek0410060] AMED-CREST, AMED [No. JP21gm1210003]). T. Tanaka was supported by the Japan Society for the Promotion of Science KAKENHI grant JP19H03708. This work was partly supported by the Uehara Memorial Foundation, the Mochida Memorial Foundation for Medical and Pharmaceutical Research, the Naito Foundation, the Mitsui Life Social Welfare Foundation, the Princes Takamatsu Cancer Research Fund, the Takeda Science Foundation, the Senshin Medical Research Foundation, the Kose Cosmetology Research Foundation, the Japan Diabetes Foundation, the Yamaguchi Endocrine Research Foundation, the Cell Science Research Foundation, the Ichiro Kanehara Foundation for the Promotion of Medical Sciences and Medical Care, the Yasuda Memorial Medical Foundation, the MSD Life Science Foundation, the Hamaguchi Foundation for the Advancement of Biochemistry, the Novartis Foundation (Japan) for the Promotion of Science and the Medical Institute of Bioregulation Kyushu University Cooperative Research Project Program.

## Author contributions

M.F., M.Y., and T.T. designed the experiments. M.F., M.Y., M.K., H.H., A.N., N.H., I.S., H.N., K. Yamagata, F.K., E.L., and R.H. performed the experiments. M.F., M.Y., M.K., I.M., A.O., K.H., K. Yokote, T.M., T.N., and T.T. analyzed the data and provided discussions. M.F., M.Y., H.H., A.O., T.M., and T.T. wrote manuscript. T.T. supervised the study.

## Competing interests

The authors declare no competing interests.
