## [Peer Review File · Nature Communications]

Liver group 2 innate lymphoid cells regulate blood glucose levels through IL-13 signaling and suppression of gluconeogenesisREVIEWER COMMENTS

Reviewer #1 (Remarks to the Author):

In this manuscript Fujimoto et al. study the role of liver ILC2s in the regulation of hepatocyte gluconeogenesis. The authors utilize state-of-the art technologies, including single-cell sequencing and epigenetics. The only conceptual criticism comes from the fact that the whole manuscript is based on a relatively artificial model: the treatment of mice with a high dose of IL33. Could the authors find a physiological model in which there is a clear expansion of ILC2s? Do the authors find evidence for a similar role of ILC2 in that model?

Technical comments:

- 1) the single-cell profile of the hepatocytes have a weird shape. Could the authors plot GLUL and CDH1? Hepatocytes should automatically order themselves in UMAPs or tSNE plots due to their strong transcriptomic zonation signature (Halpern/Iitzkovitz et al. Nature 2017). Could it be the authors still have damaged or dead hepatocytes with a poor transcriptome in their maps?
- 2) Could the authors localize the G6pc+ hepatocytes and localize the ILC2s by microscopy? Are both cells found together in the same microenvironment?

Reviewer #2 (Remarks to the Author):

- In figure 2, the authors should state which genes were used for dimensional reduction. did they ignore mitochondrial genes, cell cycle associated genes, digestion stress related genes (Heat shock protein...). Or did they just used topvariable genes ?
 - In the plot in figure 2-c, were data scaled for equal amount of cells in each of the 4 sample ? Because it is unlikely that they have equal amount of cells in the 4 groups.
 - In line 688, they write using a resolution of 5 for the clustering, which is an extremely high number, It is worth double checking this value in the script.
 - Expression plot of Gata3 in 2-d is not very convincing, are all cells supposed to be positive ? maybe the scale should start at 0 instead of 1.
 - the dot plot in fig 2-b shows strong expression of Cd3g / TCRgC1 in all the clusters. Are these expression expected on ILC2 populations ?
- In figure 4b-c, the ring shape of the population could be a sign of cell cycle bias. The authors should check cell cycle score and phase, and either display it or regress it.
- In 6a, all the genes from this family are genes known to very rapidly disregulated by digestion. Lots of publication put them in the category of digestion stress related genes, which can be avoided by cold digestion of the tissue or using transcription inhibitor during the process. In the context of looking specifically at these genes, the authors should consider or discuss this digestion stress response.
- In extended data 2-a, the authors should consider a way of displaying cluster name, more friendly for colorblind people. There is absolutely no way of guessing which population is which one (i.e, like in extended 4-a, where clusters are annotated).

Reviewer #3 (Remarks to the Author):

Fujimo et al. reported that liver group 2 innate lymphoid cells (ILC2s) could suppress the gluconeogenesis of hepatocytes thereby preventing blood glucose accumulation and insulin resistance. Mechanistically, they showed that rIL33 which could activate ILC2s to prevent elevation of blood glucose in immune competent mice. rIL33 promotes the ILC2 secretion of IL13 which suppresses transcriptional program of gluconeogenesis genes in hepatocytes in a paracrine manner.

Specifically, they began their study showing that rIL-33 could suppress blood glucose accumulation in immune competent mice but not NSG immune deficient mice. Adoptive transfer of liver ILCs could reduce glucose level in NSG mice. scRNA seq comparison of lung and liver ILC2 demonstrated that IL33 was able to induce IL13 in liver ILC2 specifically. Interestingly, rIL-33 was later shown to be able to dampen blood glucose level in IL13 +/- mice but not in IL13 -/- mice. RNA-seq analysis on hepatocytes treated with rIL13 or cAMP/glucagon showed that IL-13 suppressed gluconeogenic genes G6pc. Another scRNA seq analysis was performed comparing WT mice treated with or without rIL-33 confirmed that gluconeogenic gene G6pc was repressed. GATA3 has been reported as the central mediator of cytokine expression in ILC2; therefore, authors performed IP experiment to pull down GATA3 followed by mass spectrometry study to identify the unknown interacting partners. JUN in the AP-1 complex was identified as interacting partners of GATA3. ATAC seq analysis showed the accessibility of IL13 was increased under the influence of rIL33. They demonstrated that GATA3 binds to IL13 to activate its transcription in ILC2s. Later it was shown that AP-1 binds to GATA3 to suppress IL13 production. In liver ILC2 subsets, AP-1 level is decreased therefore unleashing IL13 transcription leading to the paracrine signaling in hepatocytes to inhibit gluconeogenesis gene transcription through STAT3.

Taken together, there are many sequencing data analyses in this study. The bioinformatics analysis is quite interesting. However, the data are quite fragmented without much experimental validations to confirm the proposed mechanisms. Conceptually, the proposed paracrine signaling is interesting. However, it is unclear how IL13 signal hepatocytes and why STAT3 represses transcription of gluconeogenesis. Also, why AP1 and STAT3 act as transcriptional repressors of IL13 and G6pc (and other gluconeogenesis genes) remain unclear when they are reported as transcriptional activators. While all the sequencing data look interesting, substantial mechanistic details have to be demonstrated experimentally.

Major concerns:

1. Paracrine signaling between ILC2 population and hepatocytes is interesting. Interaction of ILC2s and hepatocytes has not been examined experimentally. For example, co-culture experiments could be performed to study how rIL-33-stimulated ILC2 could affect hepatocytes metabolic functions
2. Metabolic readout has not been tested except blood glucose measurement in mice. How did the proposed paracrine signaling affect the metabolic output of the hepatocytes? Metabolic tracing studies on the hepatocytes after the co-culturing experiments could lead to a better understanding of the metabolic direction (flux) of the pathways influenced by G6pc.
3. Figure 1. Why nude mice that lack T cells are responsive to rIL-33?
4. Figure 3. Why only IL13 +/- mice are used as control. WT mice should be included as control as well.
5. Figure 4. It is unclear to me why ChIP assay of STAT3 is performed to evaluate its binding of Hnf4a but not other gluconeogenesis genes
6. How IL13 mediates STAT3 to repress transcription of genes in hepatocytes?
7. Figure 6. Not sure why trajectory analysis is required to show that AP-1 is associated with reduced IL13 expression. This could be directly tested out experimentally by knocking down/out AP-1 in ILC2.
8. How exactly could IL3 secreted by ILC2 affect the hepatocytes. It was proposed that the hepatocytes expressed specific receptors for IL3. This has to be functionally characterized as the paracrine signaling is an important point of this study. This interaction could not be ignored. IL3ra1, IL4ra were proposed but not validated.
9. Stat3 activation leads to Hnf4a downregulation. It is unclear how Stat3 acts as a repressor? This part is a bit confusing to me.
10. AP-1 is transcription activator. What data have led to the conclusion or idea that this transcription factor could repress IL13 expression? This needs to be explained well in the paper supported by sufficient data.
11. It is unclear why AP1 is reduced in IL-33-stimulated ILC2 subset leading to reduction of overall production of IL13.
12. The interaction of AP1/GATA3 on IL13 on ILC2 needs to be confirmed by additional experiments such as IP/Co-IP and reporter assays and independent expression assays (mRNA and ELISA) on ILC2.
13. Why throughout the manuscript only Gp6c/Pck1 is studied? There should be more gluconeogenesis genes that could be tested.

Responses to reviewers

We are very grateful for the interest and the thoughtful questions from the reviewers on our manuscript. We feel that the changes included in the revised manuscript in response to these recommendations have made the manuscript stronger, and we are grateful for the encouragement to include these analyses and additional experiments. We hope that the reviewers agree. In summary, these are the new experiments and analyses added to the paper in response to the reviewers' requests:

Fig. 3b, d, e, g, h, i, j, k, l
Fig. 4c (left panel), 4f, 4l (right panel), 4m
Fig. 5f
Fig. 6b, c
Extended Data Fig. 1d
Extended Data Fig. 2a-h
Extended Data Fig. 4a, b
Extended Data Fig. 6c
Extended Data Fig. 7e, f

We briefly show the correspondence between the new figures and answers to the major suggestions below, according to the comments of editors in the last decision letter. "In particular, we recommend you focus on the following issues:

1. Physiologically relevant model in which gluconeogenesis and secretion of IL-13 were altered
2. Answers to all queries regarding the scRNA-seq data
3. Full metabolic profiling
4. Mechanistic validation of the paracrine loop in vitro
5. Protein-protein interaction analysis to show the co-binding of factors to promoters
6. Additional target genes for the ChIP assay

1. Physiologically relevant model in which gluconeogenesis and secretion of IL-13 were altered
Extended Data Fig. 2a-h
2. Answers to all queries regarding the scRNA-seq data
Extended Data Fig. 4a, b
Extended Data Fig. 6c
Extended Data Fig. 7e, f
(Several other data are shown in the Figure for reviewers.)
3. Full metabolic profiling
Extended Data Fig. 1d
Fig. 3h, i, j, k, l
4. Mechanistic validation of the paracrine loop in vitro
Fig. 3h, i, j, k, l
Fig. 4m

5. Protein–protein interaction analysis to show the co-binding of factors to promoters
Fig. 5f
6. Additional target genes for the ChIP assay
Fig. 4l (right panel)

Moreover, additional data related to our answers are shown in the Figure for the Reviewers. It is also noted that source data are also provided as “Source Data.xlsx”.

In the following, we respond point-by-point to the reviewers’ comments.

#####

Reviewer #1 (Remarks to the Author):

In this manuscript Fujimoto et al. study the role of liver ILC2s in the regulation of hepatocyte gluconeogenesis. The authors utilize state-of-the art technologies, including single-cell sequencing and epigenetics. The only conceptual criticism comes from the fact that the whole manuscript is based on a relatively artificial model: the treatment of mice with a high dose of IL33. Could the authors find a physiological model in which there is a clear expansion of ILC2s? Do the authors find evidence for a similar role of ILC2 in that model?

Reply: First, we are delighted that you are interested in our findings and in this positive assessment of our work. We are thankful to you for raising important issues and thoughtful comments.

Thank you very much for this important suggestion. As a physiological model with altered gluconeogenesis, we examined a high-fat diet (HFD)-fed model. We observed 1-month and 3-month HFD-fed models and found clear increases in fasting blood glucose levels; blood glucose was elevated in the PTT and ITT. To evaluate ILC2 numbers and IL-13 secretion from ILC2s, we performed flow cytometry analysis. We found clear expansion of total ILC2s and IL13(+) ILC2s in liver tissue in both phases (1 month and 3 months). Chih-Hao Lee also showed that HFD-induced gluconeogenesis was exacerbated in Il13 KO BALB/c mice (Stanya et al. J Clin Invest. 2013;123(1):261-271.). In fact, we found that neutralization of IL-13 slightly but significantly increased blood glucose in the PTT in the 3-month HFD model. These data might suggest that ILC2s are physiologically increased and secrete IL-13 to compensate for excessive gluconeogenesis. We appreciate your thoughtful suggestion and giving us the opportunity to strengthen our conclusions.

Technical comments:

1) the single-cell profile of the hepatocytes have a weird shape. Could the authors plot GLUL and CDH1? Hepatocytes should automatically order themselves in UMAPs or tSNE plots due to their strong transcriptomic zonation signature (Halpern/Itzkovitz et al. Nature 2017). Could it be the authors still have damaged or dead hepatocytes with a poor transcriptome in their maps?

Thank you very much for raising this issue and providing thoughtful suggestions. In Nature 2017, Halpern et al reported that *Glul* was highly expressed near the central vein (CV) and that *Cdh1* was highly expressed in the portal node (PN). According to your advice, we observed the expression levels of *Glul* and *Cdh1* in hepatocytes and found clear gradations in their expression levels (Fig. 4c, left panel). Specifically, the Hepatocyte1 and Hepatocyte3 clusters highly expressed *Glul*, and the Hepatocyte 4 and Hepatocyte5 clusters highly expressed *Cdh1*. We believe that the hepatocytes were ordered in the UMAPs influenced by the order of the strong transcriptomic zonation signatures, as you mentioned. With regard to the weird ring shape, we rechecked the cell cycle scores and phases (G1, G2M and S) and admit that our clustering was somewhat affected by the cell cycle (Extended Data Fig. 6c). In particular, Hepatocyte2 (part of the ring shape) consisted of many S- or G2M-phase cells. We also show a plot of each cell phase and the expression of *Mki67* in the UMAPs in Fig. for Reviewers 1a-c (see below). Upon comparing the UMAPs before and after hepatocyte subclustering, we realized that additional subclustering with focusing on hepatocytes may augment the cell cycle effect. Therefore, we replaced the hepatocyte subclusters with the original clusters.

Thank you for commenting on the possibility of damaged or dead hepatocytes. We completely understand your concern because we used a much higher cutoff of mitochondrial RNA (50%) than usual for other cell types, although the same cutoff was used in another report on scRNA-seq of hepatocytes (MacParland et al. Nature Communications. 2018; 9: 4383).

Hepatocytes with a very poor transcriptome (features < 200) were already excluded in the CreateSeuratObject step in our sample, and we added a clear description on this issue in the methods. We also reviewed and displayed cell death markers and concluded that their expression was very limited (Extended Data Fig. 7e). There were few differences before and after excluding *Susd6/Fas/Bax/Bcl2*-positive cells from UMAPs (Fig. for Reviewers 1a and 1e; see below).

The most important cluster for us was Hepatocyte4 because the cells in this cluster highly expressed gluconeogenic enzymes and interacted with ILC2s via IL13 signaling. Therefore, we rechecked the traits of gene expression in Hepatocyte4. We showed the top 100 enriched GO markers of Hepatocyte4, but very few damaged or stress-related GO terms were included in the top 100 (Extended Data Fig. 7f). On the other hand, Hepatocyte4 had many metabolic process terms, including monocarboxylic acid, carbohydrate, and lipid. We also confirmed that gluconeogenic enzyme (*G6pc*, *Pck1*, and *Hnf4a*) and IL-13 receptor signaling-related genes were similarly expressed in every cell cycle phase (Fig. for Reviewers 1f; see below).

Collectively, these data suggest that the cell cycle somewhat affected the shape of the cluster, but we mitigated the effect by omitting additional subclustering. In addition, these data suggest that the cells in Hepatocyte1 are mainly located near the CV, while the cells in Hepatocyte4 and Hepatocyte5 are mainly located in the PN. We appreciate the thoughtful suggestions and feel that these analyses make our conclusions more reliable.

2) Could the authors localize the *G6pc*+ hepatocytes and localize the ILC2s by microscopy? Are both cells found together in the same microenvironment?

We are grateful for this excellent suggestion and have now carried out the appropriate immunohistochemistry. We stained G6PC first, but we did not observe sufficient staining, so we stained PCK1. Consistent with several reports, we found that PCK1 expression was stronger in the periportal region than in the central venous region (Fig. 4f). We also found that some CD3E-KLRG1+ ILC2s existed near the portal vein, which might support our idea that the G6pc/Pck1-high hepatocyte cluster interacts with hepatic ILC2s. Thank you so much for your excellent suggestion of a specific experiment to support our conclusions.

#####

Reviewer #2

(Remarks to the Author):

Reply: We are very grateful to you for the thoughtful suggestion to improve the accuracy, comprehensibility and significance of our single-cell RNA-seq analysis.

- In figure 2, the authors should state which genes were used for dimensional reduction. did they ignore mitochondrial genes, cell cycle associated genes, digestion stress related genes (Heat shock protein...). Or did they just used top variable genes ?

Thank you very much for raising this issue. We ignored the genes listed below as in a previous report and have added a list (Supplementary Table7_ignored.gene.list; Shinoda et al. PNAS. 2022. 119 (9) e2108686119). We excluded these genes because it was sometimes difficult to interpret for their biological significance, but they considerably affected clustering because of their high expression levels and expression ratios.

"mt-", "mmu", "sno", "Gm", "Rn6", "RNA", "7SK", "SNORD", "SNORA", "SCARNA", "B3g", "Vmn", "Mir", "Rik", "Snora", "Snord", "LOC", "Rn4", "OTTMUSG", "Scarna", "Rnu", "Rmr", "Rpl", "Rps", "AA", "AB", "AC", "AF", "AI", "AL", "AU", "AV", "AW", "AY", "Malat", "Tpt1", "B2m", "Actb", "Vim", "Tmsb4x", "Eef", "Fau", "BC", "B9d", "BX", "ERCC", "Hist", "Igkv", "Olf", "RP", "n-", "BB", "BY", "B4g", "CK", "CN", "CR", "CT", "Hbb", "Hba", "n-".

Mitochondria-related genes including the term "mt-" were ignored. Cell cycle-associated genes and digestion stress-related genes were included for dimensional reduction. Apparent proliferating cells were excluded before clustering of the ILC2 subset (referred to as proliferation cells in Extended Fig. 3a).

- In the plot in figure 2-c, were data scaled for equal amount of cells in each of the 4 sample ? Because it is unlikely that they have equal amounts of cells in the 4 groups.

No, they were not. We apologize for the misleading Figure and thank you for the important suggestion to make the figure more useful. Based on this question, we have made a figure with equal amounts of cells in the 4 groups and replaced it in Fig. 2c. This suggestion helped us to precisely understand the percentages of each cell type in the clusters.

- In line 688, they write using a resolution of 5 for the clustering, which is an extremely high number, it is worth double checking this value in the script.

Thank you so much for the thoughtful suggestions and we apologize for the misleading description. We double-checked our resolution setting values for the clustering in the script and would like to report that we actually used a resolution of 5.0 in the first clustering for detailed and precise cell type annotation for following reason, as mentioned in line 688, but definitely used 0.5 for ILC2 subclustering in Fig. 2a in single cell analysis of ILC2s. We have added an explanation of this point in the Methods section and apologize for the insufficient explanation.

For the 1st clustering, we used a resolution of 5.0 because there were a variety of cell types in our samples, and the cell numbers of some cell types were very small due to the sorting with ILC2 gate, as shown in Extended Fig. 3a and 3b. Since there is a previous report using the same resolution for cell type annotation to define cell type precisely (Shinoda et al. PNAS. 2022. 119 (9) e2108686119), we identified the cell type for each cluster according to this protocol.

- Expression plot of Gata3 in 2-d is not very convincing, are all cells supposed to be positive ? maybe the scale should start at 0 instead of 1.

Thank you so much for the thoughtful suggestion. We may have assumed that most ILC2s highly express Gata3 because some reports indicate that ILC2s continuously highly express Gata3 (Thomas et al. Immunity. 2012. 19;37(4):634-48.). According to your suggestion, we adjusted the scale to start at 0 instead of 1 (Fig. for Reviewer 2a; see below), which revealed that most of ILC2s still seemed to express Gata3 in our samples.

- the dot plot in fig 2-b shows strong expression of Cd3g / TCRgC1 in all the clusters. Are these expressions expected on ILC2 populations ?

Thank you very much for raising this important issue. As shown in Fig. 2e, when we normalized the expression levels among the ILC2 subsets, the expression of Cd3g/TCRgC1 seemed relatively high even though the expression in ILC2 should be very low. Therefore, we have rechecked the dot plot of all cell types and confirmed that the expression level of Cd3g/TCRgC1 in ILC2 population was very low (Extended Data Fig. 4a). Moreover, we confirmed that the TCRg expression at protein level on ILC2s was hardly detectable when we performed flow cytometry analysis (Extended Data Fig. 4b). Therefore, we have also added an explanation so as not to mislead readers.

- In figure 4b-c, the ring shape of the population could be a sign of cell cycle bias. The authors should check cell cycle score and phase, and either display it or regress it.

Thank you very much for raising this issue and for the very important suggestion. According to this advice, we rechecked the cell cycle score and phase. We found that the gene expression level of *Mki67* was not high in all clusters (Fig. for Reviewer 1c; see below), especially the Hepatocyte2 cluster (part of the ring shape), which consisted of many S-/G2M-phase cells. We admit that our clustering was at least partially affected by the cell cycle; therefore, we have displayed it in Extended Data Fig. 6c. Upon comparing the UMAPs before and after hepatocyte

subclustering, we thought additional subclustering only hepatocytes may augment the cell cycle effect. Therefore, we replaced the hepatocyte subcluster with the original clusters. We also confirmed that Hepatocyte4 expressed *G6pc*, *Pck1*, *Hnf4a*, IL13 receptors and *Stat3* independent of the cell cycle (Fig. for Reviewers 1f in red box; see below).

- In 6a, all the genes from this family are genes known to very rapidly dysregulated by digestion. Lots of publication put them in the category of digestion stress related genes, which can be avoided by cold digestion of the tissue or using transcription inhibitor during the process. In the context of looking specifically at these genes, the authors should consider or discuss this digestion stress response.

We are very grateful for the quite important question and suggestion. As you mentioned, many publications put AP-1 family members in the category of digestion stress-related genes (Brink et al. Nature Methods. 2017. 14, 935–936). Despite this fact, we consider the change and biological function of the AP-1 family to be true at least in our system for the following reasons:

1. The effect of digestion stress seemed to be limited because the expression of mitochondrial RNA was very low in all 4 ILC2 samples (Fig. for Reviewers 2b; see below).
2. Each organ was digested under exactly the same conditions among the 4 experimental groups, and thereby the effects of digestion stress were quite similar in each organ. In fact, we compared the expression of digestion-related genes reported in Nature Methods (Brink et al. Nature Methods. 2017. 14, 935–936) and found few differences between the control liver and IL-33-treated liver groups (Fig. for Reviewers 2c; see below). Despite this, we observed an apparent decrease in AP-1 genes, particularly *Junb*, *Jund*, *Jun*, *Fos*, and *Fosb* in IL-33-treated liver ILC2s compared with control livers (Fig. for Reviewers 2c; see below).
3. We also rechecked the clustering while ignoring these digestion stress genes and still found a clear negative correlation of *Il13* expression and the expression of some AP-1 family members (Fig. for Reviewers 2d-f; see below).
4. GATA3-IP-MS/MS analysis was performed using cultured ILC2s that were not affected by digestion stress (Fig. 5a, b, f). Direct GATA3-JunB binding was also confirmed by IP immunoblotting (Fig. 5f).
5. Our *in vitro* functional study (knockdown and overexpression) suggested that at least some AP-1 family members (JunB Fos and Fosb) suppress *Il13* expression in ILC2s (Fig. 5h, i, 6b, c).

Based on these data, we have added a discussion regarding the digestion stress response. We truly appreciate the questioning of this very important point.

- In extended data 2-a, the authors should consider a way of displaying cluster name, more friendly for colorblind people. There is absolutely no way of guessing which population is which one (i.e, like in extended 4-a, where clusters are annotated).

Thank you so much for the thoughtful suggestion. We have labeled the cluster in Extended Data Fig. 3a.

#####

Reviewer #3

(Remarks to the Author):

Fujimoto et al. reported that liver group 2 innate lymphoid cells (ILC2s) could suppress the gluconeogenesis of hepatocytes thereby preventing blood glucose accumulation and insulin resistance. Mechanistically, they showed that rIL33 which could activate ILC2s to prevent elevation of blood glucose in immune competent mice. rIL33 promotes the ILC2 secretion of IL13 which suppresses transcriptional program of gluconeogenesis genes in hepatocytes in a paracrine manner.

Specifically, they began their study showing that rIL-33 could suppress blood glucose accumulation in immune competent mice but not NSG immune deficient mice. Adoptive transfer of liver ILCs could reduce glucose level in NSG mice. scRNA seq comparison of lung and liver ILC2 demonstrated that IL33 was able to induce IL13 in liver ILC2 specifically. Interestingly, rIL-33 was later shown to be able to dampen blood glucose level in Il13 +/- mice but not in Il13 -/- mice. RNA-seq analysis on hepatocytes treated with rIL13 or cAMP/glucagon showed that IL-13 suppressed gluconeogenic genes G6pc. Another scRNA seq analysis was performed comparing WT mice treated with or without rIL-33 confirmed that gluconeogenic gene G6pc was repressed. GATA3 has been reported as the central mediator of cytokine expression in ILC2; therefore, authors performed IP experiment to pull down GATA3 followed by mass spectrometry study to identify the unknown interacting partners. JUN in the AP-1 complex was identified as interacting partners of GATA3. ATAC seq analysis showed the accessibility of IL13 was increased under the influence of rIL33. They demonstrated that GATA3 binds to IL13 to activate its transcription in ILC2s. Later it was shown that AP-1 binds to GATA3 to suppress IL13 production. In liver ILC2 subsets, AP-1 level is decreased therefore unleashing IL13 transcription leading to the paracrine signaling in hepatocytes to inhibit gluconeogenesis gene transcription through STAT3.

Taken together, there are many sequencing data analyses in this study. The bioinformatics analysis is quite interesting. However, the data are quite fragmented without much experimental validations to confirm the proposed mechanisms. Conceptually, the proposed paracrine signaling is interesting. However, it is unclear how IL13 signal hepatocytes and why STAT3 represses transcription of gluconeogenesis. Also, why AP1 and STAT3 act as transcriptional repressors of IL13 and G6pc (and other gluconeogenesis genes) remain unclear when they are reported as transcriptional activators. While all the sequencing data look interesting, substantial mechanistic details have to be demonstrated experimentally.

Reply: We are very grateful to you for appreciating our work and raising many important issues and thoughtful specific suggestions.

Major concerns:

1. Paracrine signaling between ILC2 population and hepatocytes is interesting. Interaction of ILC2s and hepatocytes has not been examined experimentally.

For example, co-culture experiments could be performed to study how rIL-33-stimulated ILC2 could affect hepatocytes metabolic functions

We truly appreciate the specific suggestion to strengthen our conclusions and performed coculture experiments of primary hepatocytes and rIL-33-stimulated hepatic ILC2s (Fig. 3h). After 6 h of coculture, we performed qPCR of hepatocytes, which revealed that coculture decreased the expression of gluconeogenesis-related genes (*G6pc*, *Pck1*, *Hnf4a*, *Fbp1*, *Pcx*, *Ppargc1a*, *Pfkl* and *Pklr*); however, this effect was abolished by neutralization of IL-13 (Fig. 3i). These data were consistent with the decreased gluconeogenesis in liver tissue after rIL-33 treatment via paracrine signaling between ILC2s and hepatocytes. Thanks to your valuable advice, we feel that our conclusion is supported by data on the direct interaction of hepatocytes and ILC2s.

2. Metabolic readout has not been tested except blood glucose measurement in mice. How did the proposed paracrine signaling affect the metabolic output of the hepatocytes? Metabolic tracing studies on the hepatocytes after the co-culturing experiments could lead to a better understanding of the metabolic direction (flux) of the pathways influenced by G6pc.

Thank you so much for the additional specific suggestion on the coculture experiment. According to this suggestion, we performed metabolome analysis on hepatocytes after coculture with ILC2s (Fig. 3h). We found changes in metabolites after coculture in line with our notion (Fig. 3j, k, l).

Fig. 3j:

For amino acids, four amino acids were significantly changed by ILC2 coculture. Interestingly, three of the four significant amino acids belonged to the pyruvic acid pathway (Ala, Cys, and Gly). These three amino acids are classified into glucogenic amino acids, which can be substrates for gluconeogenesis (Andrea et al. Biochemical Education. 2000. 28, 27-28, <https://onlinelibrary.wiley.com/doi/pdf/10.1111/j.1539-3429.2000.tb00007.x>).

Fig. 3k:

Regarding the glycolysis/gluconeogenesis pathway, F1,6P levels were significantly increased in hepatocytes cocultured with ILC2s, which is consistent with the decreased expression of *Fbp1* in hepatocytes cocultured with ILC2s. In addition, phosphoenolpyruvate (PEP) levels were also significantly decreased by coculture, which is also consistent with the decreased expression of *Pck1*. G6P was not significantly altered by coculture. The glucose-rich medium might have influenced the content of G6P in hepatocytes.

Fig. 3l:

Regarding the tricarboxylic acid (TCA) cycle, clear increases in citric acid and malic acid were observed. While the exact mechanisms and biological meanings of the accumulation of these metabolites are not fully understood, ILC2 coculture might have also affected the TCA cycle in hepatocytes as a consequence in gluconeogenic pathway.

We feel that these results, taken together, support the idea that hepatic ILC2s directly affect gluconeogenesis in hepatocytes and truly appreciate your thoughtful suggestion.

3. Figure 1. Why nude mice that lack T cells are responsive to rIL-33?

Thank you so much for raising this issue. We were also surprised that the blood glucose-lowering effect of rIL-33 was not weakened at all in T-cell-deficient nude mice. For this reason, we confirmed that liver ILC2s were still present in nude mice (Fig. for Reviewers 3a; see below). In addition, the expression level of *Iil3* in liver tissue was still high in nude mice and was actually higher than that in wild-type mouse livers in response to IL-33 stimulation, which was confirmed with additional samples (Fig. 3b, Fig. for Reviewers 3b, same dataset; see below). These data might support our hypothesis and the importance of hepatic ILC2s as sources of IL-13. It is noted that nude mice lack T cells but ILC2s are present, whereas NSG mice lack both T cells and ILC2s.

4. Figure 3. Why only *Iil3*^{+/-} mice are used as control. WT mice should be included as control as well.

Thank you so much for the suggestion. We have included data from wt mice (Fig 3d, 3e, 3g). The results show that wild-type mice have a trend similar to that of *Iil3*^{+/-} mice.

5. Figure 4. It is unclear to me why ChIP assay of STAT3 is performed to evaluate its binding of *Hnf4a* but not other gluconeogenesis genes

Thank you for raising this point, and we apologize for the insufficient explanation. As you might have thought, the ChIP assay of STAT3 to evaluate its binding to promoters of *G6pc* or *Pck1* was more understandable than that of *Hnf4a*. To make our logic clearer, we have reordered the figures and results. We first show the results for STAT3-ChIP-PCR upstream of *G6pc* (Fig. 4l left panel). Additionally, we performed an additional ChIP assay of STAT3 for the upstream of *Pck1* and confirmed the association of STAT3 with its upstream region (Fig. 4l right panel).

We have placed the results related to *Hnf4a* in Extended Data Fig. 7 a-d and explain them one by one in the text. *Hnf4a* was the most enriched gene upstream of the DEGs between the PBS and IL-33 groups, and its downstream genes were highly expressed in Hepatocyte4 (Extended Data Fig. 7a). IL-33 treatment significantly decreased *Hnf4a* expression in Hepatocyte4, in which *Hnf4a* and gluconeogenic genes were highly expressed (Extended Data Fig. 7b). From these data, we hypothesized that STAT3 binds to *Hnf4a* upstream, and motif analysis using JASPAR showed a significantly enriched STAT3 motif upstream of *Hnf4a* (Extended Data Fig. 7c). In addition, we found that rIL-13 treatment promoted STAT3 binding upstream of *Hnf4a* in primary hepatocytes (Extended Data Fig. 7d). These data suggest that STAT3 also binds upstream of *Hnf4a* and is involved in the suppression of *Hnf4a* expression.

We appreciate the opportunity to improve our logic and discussion.

6. How does IL13 mediates STAT3 to repress transcription of genes in hepatocytes?

Thank you very much for raising this issue. It would be of interest to determine the molecular mechanism by which IL13 causes STAT3-dependent transcriptional repression of glycogenic genes in hepatocytes. Although to the best of our knowledge, this molecular mechanism has not been fully elucidated, we presume that IL-13 promotes STAT3 phosphorylation and STAT3 binding upstream of genes, leading to suppression of gene expression, as reported previously (Stanya et al. J Clin Invest. 2013;123(1):261-271.). Indeed, the results of STAT3-ChIP assays showed its associations with upstream regions of several genes, including *G6pc*, *Pck1* and *Hnf4a* (Fig. 4l, Extended Data Fig. 7d). The mechanisms by which STAT3-dependent transcriptional activation and repressive function are differentially used are also not fully understood. Nevertheless, further investigation will be required to elucidate underlying mechanisms of STAT3-dependent transcriptional repression of glycogenic genes in hepatocytes, although we think that this issue is not the main topic of this paper.

7. Figure 6. Not sure why trajectory analysis is required to show that AP-1 is associated with reduced IL13 expression. This could be directly tested out experimentally by knocking down/out AP-1 in ILC2.

Thank you for this important suggestion. According to this suggestion, we performed a knockdown experiment on hepatic ILC2s for additional AP-1 family members (*Junb* and *Jund* in Fig. 5i and *Jun*, *Fos*, *Fosb* and *Fosl2* in Fig. 6b). Knocking down *Junb*, *Fos*, and *Fosb* increased the expression of *Il13*. In addition, we also overexpressed *Junb* in primary liver ILC2s or in the ILC2/b6 cell line and found that *Junb* overexpression significantly suppressed *Il13* expression on ILC2s (Fig. 6c).

We agree that trajectory analysis is typically useful for models of differentiation. Although the application of trajectory analysis for a purpose such as this case may not be typical, we included this analysis as a supplemental model to illustrate the transition of ILC2 maturation in response to IL-33 and to provide data figure for changes in Gata3 downstream genes including *Il13* and AP-1 family members in single cell signature. We appreciate if reviewers can agree that this result would be informative.

8. How exactly could IL13 secreted by ILC2 affect the hepatocytes. It was proposed that the hepatocytes expressed specific receptors for IL13. This has to be functionally characterized as the paracrine signaling is an important point of this study. This interaction could not be ignored. *Il13ra1*, *Il4ra* were proposed but not validated.

Thank you very much for raising this important point. According to this, we performed an experiment with knockdown of *Il13ra1*, *Il4ra*, *Stat3*, and *Stat6* in hepatocytes and evaluated the effect of rIL-13 on gluconeogenic genes. As expected, knockdown of *Il13ra1*, *Il4ra*, and *Stat3*, but not *Stat6*, significantly increased *G6pc* or *Pck1* expression in hepatocytes cultured with IL-13. These results suggest that the *Il13ra1/Il4ra* heterodimer and subsequent *Stat3* signaling mediate the suppression of *G6pc* and *Pck1* expression by IL-13 (Fig. 4m). We are grateful for the specific suggestion to validate the concept we proposed.

9. *Stat3* activation leads to *Hnf4a* downregulation. It is unclear how *Stat3* acts as a repressor? This part is a bit confusing to me.

Thank you for raising this issue. As mentioned above in question 6, we were also interested in this point but could not clearly explain it. However, several reports support the suppressive function of STAT3 in IL-13 signaling (Stanya et al. *J Clin Invest.* 2013;123(1):261-271) and IL-6 signaling (Ramadoss et al. *Mol Endocrinol.* 2009. 23(6):827-37). It has also been reported that other transcription factors function as both activators and repressors (Shaulian et al. *Oncogene.* 2001. 20, 2390–2400, Mittelstadt et al. 2012. *PLoS ONE* 7(8): e42152). It would be of interest to reveal the molecular mechanism of STAT3-dependent transcriptional repression in hepatocytes; however, further study is needed to understand this mechanism.

10. AP-1 is transcription activator. What data have led to the conclusion or idea that this transcription factor could repress *Il13* expression? This needs to be explained well in the paper supported by sufficient data.

Thank you so much for this important suggestion. As you mentioned, AP-1 is a well-known activator. However, in the context of cancer, it has been reported that AP-1 can suppress tumor suppressor genes or matrix metalloproteases (Shaulian et al. *Oncogene.* 2001. 20, 2390–2400, Mittelstadt et al. 2012. *PLoS ONE* 7(8): e42152). Additionally, it has been reported that AP-1 binds to glucocorticoid receptor (GR) and antagonizes the transcriptional function of GR (Shule et al. *Cell.* 1990. 62, 1217-1226.; [https://www.cell.com/cell/pdf/0092-8674\(90\)90397-W.pdf](https://www.cell.com/cell/pdf/0092-8674(90)90397-W.pdf)). Because the expression of some AP-1 members was apparently decreased in IL-33-activated hepatic ILC2s, we considered that AP-1 might act as a suppressor in hepatic ILC2s.

We agree that functional assays are needed to support this point. We consider our most reliable functional assay results to be the following:

1. Knockdown of *Junb*, *Fos*, and *Fosb* enhanced *Il13* expression on ILC2s (Fig. 5i, 6b).
2. *Junb* overexpression decreased *Il13* expression both in primary ILC2s and in an ILC2 cell line (Fig. 6c).

We have added an explanation of this point in discussion section and appreciate the opportunity to improve our discussion.

11. It is unclear why AP1 is reduced in IL-33-stimulated ILC2 subset leading to reduction of overall production of *Il13*.

Thank you for raising this issue; we are also interested in this point. Frankly, we do not have a clear answer at this point, but we suppose that the mechanism is not simply explained by the direct effects of ILC2s. Changes in the AP-1 family were different between liver ILC2s and lung ILC2s, suggesting that the niche environment and its differences in distinct organ may be involved in this AP-1 reduction. In fact, the DEGs between IL-33-stimulated liver ILC2s and lung ILC2s showed clear differences in the expressions of AP-1 genes (*Fosb* and *Fos*) and different traits of ILC2s in different organs (Fig. for Reviewers 3c, d; see below). Further investigation is needed to reveal this mechanism.

12. The interaction of AP1/GATA3 on IL13 on ILC2 needs to be confirmed by additional experiments such as IP/Co-IP and reporter assays and independent expression assays (mRNA and ELISA) on ILC2.

Thank you so much for the important suggestions. We first performed GATA3-IP/JunB western blotting (WB) using primary hepatic ILC2s, but it was difficult to detect enough signal, probably because of the limited sensitivity of IP-WB. Then, we used the ILC2 cell line transfected with a Flag-tagged GATA3/JunB-overexpressing retrovirus. We performed GATA3(FLAG)-IP/JunB-WB using this cell line and observed a clear GATA3-JunB interaction on ILC2s (Fig. 5f). We also observed that *Junb* overexpression suppressed *Il13* expression in both primary ILC2s and an ILC2 cell line. We appreciate your valuable suggestions.

13. Why throughout the manuscript only Gp6c/Pck1 is studied? There should be more gluconeogenesis genes that could be tested.

Thank you for the thoughtful suggestion. According to this, we performed qPCR for additional gluconeogenesis-related genes (*G6pc*, *Pck1*, *Fbp1*, *Pcx*, *Pparg1c*, and *Hnf4a*) in an *in vivo* liver tissue model. Consistent with the *in vivo* data shown in Fig. 1 (Fig. 1c, d, e), the effect of rIL-33 on gluconeogenesis genes was confirmed in both wild-type and nude mice (Extended Data Fig. 1d). The effect of rIL-33 on these genes disappeared in NSG mice, which suggests that this effect is mediated by the hepatic ILC2-IL13 axis. We also included the same gene sets in an evaluation of hepatocytes cocultured with ILC2s in order to interpret the metabolic processes more precisely. We appreciate the important suggestions, which helped us to obtain more information from our existing experimental systems.

We hope that these responses have satisfactorily addressed the thoughtful comments of the reviewers. We are very grateful for their many points of encouragement to improve this paper and are delighted that the new experiments and analyses have strengthened our conclusions. We hope that they and the editors will now find this work acceptable for publication in *Nature Communications*.

Figure for Reviewers 1

Figure for Reviewer 1

a, Hepatocyte-enriched cells isolated by collagenase perfusion via the portal vein. scRNA-seq data ($n = 11,327$ single cells from the liver) for phosphate-buffered saline- and rIL-33-treated livers, shown as nonlinear representations of the top 50 principal components; cells are colored based on UMAP-based cell cluster (same data as Figure 4b). **b**, The cell phase of each cell is shown in one plot (left) or split plot (right). **c**, UMAP plots showing *Mki67* expression in all cells. **d**, UMAP plots showing *Susd6*, *Fas*, *Bax* and *Bcl2* expression in all cells. **e**, UMAP plots showing only cells without expression of any of *Susd6*, *Fas*, *Bax* or *Bcl2*. **f**, Dot plot showing the expression of gluconeogenesis-related genes (*G6pc*, *Pck1*, and *Hnf4a*) and IL-13 signaling pathway genes (*Il13ra1*, *Il4ra*, *Stat3*, and *Stat6*) in each cluster at different cell cycle phases (S, G1, and G2M).

Figure for Reviewers 2

Figure for Reviewer 2

a, UMAP plots showing the expression of Gata3 with a scale starting at 0. **b**, nFeatures_RNA, nCount_RNA and percent of mitochondrial RNA in 4 samples each. **c**, Average gene expression of digestion stress-related genes in the AP-1 family or other than the AP-1 family on ILC2s in the liver (left) and lung (right). **d**, **e**, Cells were clustered, ignoring stress-related genes, including heat shock proteins, other than the AP-1 family. UMAP plots of liver (**d**) and lung (**e**) ILC2s colored by UMAP clusters (left) or treatment (right). **f**, Correlation matrix for AP-1 family members and Gata3 downstream genes based on the average expression of the clusters shown in **d** and **e**.

Figure for Reviewers 3

Figure for Reviewers 3

a, Gating for Lin-Thy1+CD127+ST2+ ILC2s in livers from nude mice. **b**, *Il13* expression levels in liver tissue from IL-33-treated wild-type or nude mice (normalized to L32) (same data as Figure 3b). **c**, Heatmap showing the top 50 differentially expressed genes (DEGs) between IL-33-treated liver ILC2s and IL-33-treated lung ILC2s. **d**, Enriched GO list in DEGs between IL-33-treated liver ILC2s or IL-33-treated lung ILC2s.

REVIEWERS' COMMENTS

Reviewer #1 (Remarks to the Author):

I congratulate the authors for their manuscript and thank them for the performed experiments.

Reviewer #2 (Remarks to the Author):

I want to thank the author for discussing all the points raised during the revision.
I have no concerns and have been convinced by the author's comments.

Reviewer #3 (Remarks to the Author):

Authors have provided reasonable answers to my concerns.

Reviewer #1 (Remarks to the Author):

I congratulate the authors for their manuscript and thank them for the performed experiments.

Response to the Reviewer #1

Thank you very much for reviewing our manuscript again and we are very happy to your response. we are very grateful that you appreciated our additional experiments, their interpretation and discussion.

Reviewer #2 (Remarks to the Author):

I want to thank the author for discussing all the points raised during the revision. I have no concerns and have been convinced by the author's comments.

Response to the Reviewer #2

Thank you very much for reviewing our manuscript again and we are very happy to your response. We are very grateful that you appreciated our additional explanation, analysis and discussion.

Reviewer #3 (Remarks to the Author):

Authors have provided reasonable answers to my concerns.

Response to the Reviewer #3

Thank you very much for reviewing our manuscript again and we are very happy to your response. we are very grateful that you appreciated our additional experiments, their interpretation and discussion.